# Vegetation Influence and Environmental Controls on Greenhouse Gas Fluxes from a Drained Thermokarst Lake in the Western Canadian Arctic

June Skeeter[1], Andreas Christen[2], Andrée-Anne Laforce[3], Elyn Humphreys[3], Greg Henry[1]

[1]Department of Geography, The University of British Columbia, Vancouver, V6T1Z2, Canada
[2]Environmental Meteorology, Faculty of Environment and Natural Resources, Albert-Ludwigs Universität Freiburg, Freiburg, Germany
[3]Department of Geography and Environmental Studies, Carleton University, Ottawa, K1S5B6, Canada

*Correspondence to*: June B. Skeeter (skeeter1@mail.ubc.ca)

**Abstract.** Thermokarst features are widespread in ice-rich regions of the circumpolar Arctic. The rate of thermokarst lake formation and drainage is anticipated to accelerate as the climate warms. However, it is uncertain how these dynamic features impact the terrestrial Arctic carbon cycle. Methane ($CH_4$) and carbon dioxide ($CO_2$) fluxes were measured during peak growing season using eddy covariance and chambers at Illisarvik, a 0.16 km$^2$ thermokarst lake basin that was experimentally drained in 1978 on Richards Island, Northwest Territories, Canada. Vegetation in the

basin differs markedly from the surrounding dwarf-shrub tundra and included patches of tall shrubs, grasses and sedges with some bare ground and a small pond in the centre. During the peak growing season, temperature and wind conditions were highly variable and soil water content decreased steadily. Basin-scaled net ecosystem $CO_2$ exchange (NEE) measured by eddy covariance was -1.5 [$CI_{95\%} \pm 0.2$] g C-$CO_2$ m$^{-2}$ d$^{-1}$; NEE followed a marked diurnal pattern with no day-to-day trend during the study period. Variations in half-hourly NEE were primarily controlled by

photosynthetic photon flux density and influenced by vapor pressure deficit, volumetric water content and the presence of shrubs within the flux tower footprint, which varied with wind direction. Net methane exchange (NME) was low (8.7 [$CI_{95\%} \pm 0.4$] mg $CH_4$ m$^{-2}$ d$^{-1}$ and had little impact on the growing season carbon balance of the basin. NME displayed high spatial variability and sedge areas in the basin were the strongest source of $CH_4$ while upland areas outside the basin were a net sink. Soil moisture and temperature were the main environmental factors influencing

NME. Presently, Illisarvik is a carbon sink during the peak growing season. However, these results suggest that rates of growing season CO2 and CH4 exchange rates may change as the basin's vegetation community continues to evolve.

Keywords: Climate Change, Arctic, Permafrost, Thermokarst, Carbon Dioxide, Methane

## 1 Introduction

The northern permafrost region stores approximately 50% of global organic soil carbon in 16% of the terrestrial land area (Tarnocai et al. 2009). Thermokarst landscapes account for approximately 20% of the land area in this region and hold about half of its organic soil carbon (Olefeldt et al., 2016). Lake thermokarst landscapes are widespread in poorly drained, sedimentary permafrost lowlands with excess ground ice volume and constitute about a third of all

thermokarst area (French, 2013; Olefeldt et al., 2016). Thermokarst lakes are a prominent landscape feature of the

Western Canadian Arctic (Mackay, 1999; Marsh et al, 2009; Lantz & Turner, 2015). These lakes drain, sometimes catastrophically, forming drained thermokarst lake basins (DTLBs) via bank overflow, ice wedge erosion, coastal erosion, and stream migration (Billings and Peterson, 1980; Mackay, 1999). Lake formation and drainage is a natural part of the thaw lake cycle, but it is anticipated that climate change will accelerate or disturb this cycle, potentially altering the regional carbon balance (Jones et al., 2018).

Net ecosystem exchange (NEE), ecosystem respiration (ER) and gross primary productivity (GPP), where NEE $=$ ER $-$ GPP are lower in the Arctic than warmer regions but have significant seasonal cycles and variability between vegetation types (Virkkala et al., 2018). Future trajectories in NEE will in large part be governed by ER (Biasi et al., 2008; Cahoon et al.,2012). Dominant vegetation types in the Western Canadian Arctic are erect-shrub tundra and wetlands (Walker et al., 2005). Growing season NEE is typically negative across these units throughout the Arctic

indicating a net $CO_2$ sink as GPP exceeds ER in part due to cold and/or anoxic soil conditions (Virkkala et al., 2018; Lafleur et al., 2012). Annual NEE can be positive or negative with large variation in GPP linked to annual weather variability (Virkkala et al., 2018, McGuire et al., 2009). Arctic net methane exchange (NME) is positive because wetland areas are strong methane ($CH_4$) sources while upland areas with better drainage can be net sinks (Whalen and Reeburgh, 1990; McGuire et al., 2009; Sturtevant and Oechel, 2013).

Thermokarst lakes are well recognized sources of $CH_4$ (Walter et al., 2007) which is 28 times as potent as carbon dioxide ($CO_2$) on a 100-year time scale (IPCC, 2014). Thermokarst lake formation and expansion is expected to exert a positive feedback on climate change and accelerate Arctic warming in the near term, but modelling suggests that drainage may limit expansion and result in decreased lake area by the end of the century (van Huissteden et al., 2011). Post drainage, DTLBs undergo rapid ecological succession. In colder tundra environments, wet meadows or

polygonal landscapes dominated by sedges, grasses and rushes will form (Lara et al., 2015). In slightly warmer, boreal and transitional regions, DTLBs often become dominated by willows and other shrubs (Lantz and Turner, 2015). Carbon exchange in DTLBs of various ages has been examined in a few studies, almost exclusively focused on the Barrow Peninsula in Northern Alaska. DTLB NEE during the growing season is negative with greatest $CO_2$ uptake in younger basins and decreasing net uptake as basins age in this region (Zona et al., 2010; Zulueta et al., 2011;

Sturtevant and Oechel, 2013; Lara et al., 2015). DTLB source/sink strength of $CH_4$ was found to be highly variable depending on vegetation and ground conditions (Lara et al., 2015). NME is highest in wet meadows and remnant ponds but considerably reduced in areas with better drainage (Zona et al., 2009; Zona et al., 2012; Lara et al., 2015). There may be regional variations in the carbon balance of DTLBs. For example, a shrub dominated ancient DTLB known as Katyk in the Indigirka lowlands of Siberia shows considerably higher growing season carbon uptake than

young Alaskan DTLBs with comparable NME (van der Molen et al., 2007; Parmentier et al. 2011). Similarly, DTLBs in the Western Canadian Arctic may have different carbon fluxes than Alaskan DTLBs due to differences in climate and vegetation composition.

In this study, fluxes of $CO_2$ and $CH_4$ were measured at Illisarvik, an experimentally drained thermokarst lake basin on Richards Island in the Western Canadian Arctic, Northwest Territories, Canada. Fluxes of $CO_2$ and $CH_4$ were

measured during the peak growing season using a combination of closed chamber and eddy covariance (EC)

measurements. NEE was calculated from fluxes and storage change and was separated into ER and GPP. Here we report on: 1) the spatial and temporal variability of the NEE and NME during the growing season, 2) the vegetation and environmental factors influencing NEE and NME, 3) how the growing season carbon balance at Illisarvik compares to other DTLBs, and 4) potential future carbon balance trajectories as Illisarvik's vegetation communities continue to evolve.

## 2 Methods

### 2.1 Study Site and Data Collection

The study took place at Illisarvik, a DTLB on Richards Island (69˚28'47.5" N, 134˚35'18.7" W), that was drained experimentally in 1978 (Mackay, 1981). Illisarvik has since served as the focus of studies on permafrost growth, active layer development and vegetation succession (Ovenden, 1986; Mackay and Burn, 2002; O'Neil et al., 2012; Wilson et al., 2019). At the nearby Tuktoyaktuk climate station mean annual air temperature ($T_a$) is -10.1 °C, July is the warmest month with a mean of 11°C and January is the coldest at -27°C. Mean annual precipitation is 160.7 mm yr$^{-1}$, the majority falling as rain in the summer and autumn. Snow cover typically lasts from mid-September or early October to late May (Environment Canada. 2016). Tuktoyaktuk is 60 km east of Illisarvik and in similar proximity to the coast so the climatology is expected to be similar at Illisarvik.

In the 39 years since drainage, Illisarvik has undergone rapid vegetation succession. After drainage, there were two remnant ponds. In the first five years after drainage, vegetation colonized the basin margins and wetter areas (Ovenden, 1986). By 1999, low vegetation had proliferated across most of the basin and taller willows had become established along the basin margins (Mackay and Burn; 2002). By 2010, some of the willows had grown to be 3 m in height (O'Neil and Burn; 2012). Current vegetation at Illisarvik is diverse relative to the dwarf-shrub tundra of the surrounding uplands (Table 1); the basin hosts a mix of woody shrubs (*Salix* spp*.*, *Betula* spp*.*, & *Alnus* spp), wetland vegetation (*Carex aquatilis, Arctophila fulva*, etc.), and various grasses (*Pocacea* spp*.*) (Wilson et al. 2019). The basin is partly ringed by a terrace of peat that formed after a partial drainage event ~ 5000 years BP and supports vegetation similar to the uplands (Michel et al., 1989). An ancient DTLB is located 100 m to the south of the Illisarvik basin and the Arctic Ocean is to the west of the basin, separated by a ridge of upland tundra about 50 m wide at its narrowest (Fig 1).

A vegetation survey of species composition and abundance was done on a 50 m grid in and around the basin during the 2016 study period (Wilson et al., 2019). A vegetation map was created with ten units based on plant functional type and vegetation structure, with sub-units denoting sub-canopy vegetation. The unit boundaries between grid points were estimated visually by traversing the grid lines. Additional survey data on vegetation units and canopy height were collected manually with a GPS in the proximity of the EC station because greater resolution was needed for footprint modelling. Aerial imagery was collected on July 23$^{rd}$ over two flights using a Phantom 2 drone (DJI, Shenzhen, China). The GPS points and drone imagery were used to cross reference and modify the map of Wilson et al. (2019). The ten units were then aggregated into six broader surface cover classes (listed from largest to smallest

areal fraction within the footprint climatology ($F_{Clim}$) see Section 2.3 for definition): shrub, grass, sedge, upland, sparse, and water classes (Fig 1 & Table 1).

**2.2 Weather and Soil Measurements**

Weather data were logged on a CR1000 datalogger (Campbell Scientific Inc, Logan, UT, USA; CSI) at 5-minute intervals. Net all-wave radiation ($R_n$) and photosynthetic photon flux density ($PPFD$) were measured with a NRLite

net radiometer (Kipp & Zonen, Delft, Netherlands) and a SQ-110 quantum sensor (Apogee Instruments, Logan, UT, USA), respectively 3.2 m above the grass surface on the main EC system tripod (Fig 1). A shielded HMP35 (CSI) recorded $T_a$ and relative humidity ($RH$) 2 m above the surface. A tipping bucket rain gauge (R.M Young Company, Travers City, MI, USA) was placed 3 m to the west of the main tripod. Soil temperature and moisture were measured within soil pits in two different vegetation types near the tripod: Grass (30 m to the east) and Shrub (40 m to the north).

Measurements were made of ground heat flux ($G$) with custom-made heat flux plates, soil temperatures ($T_s$) with custom type-T thermocouples at depths of 0.08 m, and 0-20 cm integrated volumetric water content (VWC) with CS616 water content reflectometers (CSI). The soil measurements were recorded at 30-minute intervals on CR10x dataloggers (CSI). The climate and soil stations operated uninterrupted from July 10[th] (day 192) and July 11[th] (day 193), respectively, until August 7[th], 2016 (day 220). On July 11[th] and August 6[th] thaw depth was measured at each of

the 10 chamber sites (see below). Thaw depth was measured by inserting a graduated steel probe into the ground to point of refusal. Each site was probed five times: the median value has been used as the thaw depth at each location. On July 12th and 15th, a large herd of reindeer (500 + animals) visited Illisarvik. They mostly avoided the tripod but did graze near it for about an hour on July 12th which may have affected greenhouse gas fluxes.

**2.3 EC Fluxes**

An EC system was placed in the southwestern portion of the basin (69° 28' 47.82", -134° 35' 18.6") and measured fluxes of $CO_2$ ($F_{CO2}$) and $CH_4$ ($F_{CH4}$) for the full study period between July 10[th] and Aug 7[th], 2016. The EC system consisted of an open-path infrared $CO_2/H_2O$ gas analyser (IRGA) (model LI-7500, LI-COR Inc., Lincoln, NK, USA; LI-COR), an open-path $CH_4$ analyser (model LI-7700, LI-COR) and a CSAT3 sonic anemometer (CSI) mounted on a tripod at a measurement height ($z_m$) of 3 m (Fig 2). The EC data and air pressure ($P_a$) were logged at 10 Hz on the

LI-7550 Analyzer Interface Unit (LI-COR). The CSAT3 was oriented to the northeast (40°) because climatology for Tuktoyaktuk indicated northerly and easterly winds are typical for July and August (Environment Canada, 2016). Half-hourly fluxes were calculated with EddyPro V.6.2.0 (LI-COR). The software performed statistical assessments (Vickers and Mart, 1997), low and high frequency spectral corrections (Moncrieff et al., 1997 and 2004), a double rotation (Wilczak et al., 2001), applied the WPL correction to account for density fluctuations (Webb et al., 1980),

and computed quality control (qc) flags (Mauder and Foken, 2004). Post processing treatments included: storage correction (calculating the net flux as the sum of the observed scalar flux and the rate of change in scalar concentration at $z_m$), filtering fluxes by friction velocities ($u_*$) below 0.1 m s[-1], removing qc flags = 2 (Mauder and Foken, 2004), and the mean absolute deviation spike removal algorithm (Papale et al., 2006). Additionally, observations with mean winds from 220° ± 30° were removed to avoid uncertainties associated with the wake of the sonic anemometer, and

observations were removed during precipitation events and when the open-path analysers indicated there were any other obstructions within the path (Aubinet et al., 2012).  The data were gap-filled using neural networks (NN) which have been applied to $F_{CO2}$ and $F_{CH4}$ in other studies (Moffat et al., 2010; Dengel et al., 2013).  Details of the NN methodology are described in Appendix A.

The flux footprint represents the influence of upwind areas on a measured scalar flux and the footprint climatology is

the average of individual footprints over a time period. Evaluation of the flux footprints and climatology help evaluate the reliability of the dataset and estimate the source area of each individual half-hourly EC flux measurement.  A scalar flux $F_c$ sampled at $(0,0,z_m)$, where $z_m$ is the height of the EC instrumentation, can be represented as the integral of the flux footprint function $f(x,y)$ and the distribution of sources/sinks ($Q_c$) over a domain $D$ (Kljun et al., 2015):

$$F_c(0,0,z_m) = \int_D Q_c(x,y) f(x,y) \tag{1}$$

The flux contribution of upwind source areas increases sharply upwind from the measurement location to a peak then decrease gradually with increasing distance (Schmid, 2002).  The empirically derived flux footprint function of Kljun et al. (2015) was used to estimate the source area of each half hourly flux measurement.

The model requires boundary layer heights which were not measured onsite.  Half hourly boundary layer heights were interpolated from three-hour estimates obtained from the Global Data Assimilation System of the U.S. National

Oceanic and Atmospheric Administration.  The model also requires the aerodynamic roughness length ($z_0$) which is influenced by the canopy height and spacing.  Canopy height ($C_h$) varied considerably within the basin (from >1 m in the north to ~0 m in the bare ground areas).  Canopy height variability was lower in the vicinity of the EC tripod but ranged from 0.35- 0.55 m with a few taller shrubs approaching 1 m.  Median $z_0$ was calculated for 30˚ wind sectors following Paul-Limoges et al. (2013).  This calculation was performed for near neutral conditions $-0.05 \leq \frac{zm}{L} \leq$

$0.05$, where $L$ is the Obukhov length.  The $z_0$ for each wind sector was found to be insensitive to zero-plane displacement height, $d$, as $z_m \gg d$, so the mean value of $d$ around the tripod was used, where $d = 2/3\ C_h$. Zero-plane displacement did not change significantly over the course of the study so $z_0$ remained fixed over the study period for each wind sector.

For each half-hourly flux observation, $f(x,y)_i$ was solved at 1 m² resolution over a 1 km² domain centred on the EC

tripod.  Then, $f(x,y)_i$ were intersected with the surface classes to determine the relative contribution of each surface type to each flux observation (referred to as $F_{Shrub}$, $F_{Sedge}$, etc.).  The footprint function is technically infinite so a fraction of each $f(x,y)_i$ was not contained within the model domain.  The out-of-domain source fraction ranged from 1.8% to 4.9% with a mean of 3.2% and was assumed to have minimal impact on the analysis.   The flux footprint climatology ($F_{Clim}$) was calculated by averaging the half hourly flux footprints over the study period and is shown in

Figure 1. Table 2 shows the flux contribution of each vegetation class.

**2.4 Closed Chamber Measurements**

In addition to EC measurements, fluxes of $CO_2$ and $CH_4$ were sampled using a static non-steady state chamber flux technique on 11 dates between July 12 and August 5, 2016 (Laforce, 2018).  Nineteen chamber collars were located

at ten sites, eight sites within and two outside the basin (Fig 1). Each surface cover class was represented by at least

one chamber site, except for open water. At each vegetated site a pair of collars were installed 20 cm apart, except at the 'sparse' site where only one collar was installed. The above ground biomass was removed from one of the collars at each vegetated site. There were three replicates (six collars) for the Shrub class, two for the Sedge, Grass, and Upland tundra, and no replicates for the Sparse class. PVC collars 30 cm long and 24.3 cm in diameter were inserted to a depth of approximately 15 cm. The chambers were 34 cm tall and made out of polycarbonate covered in black

opaque tape to maintain dark conditions inside the chamber (for more details, see Martin et al., 2018). The chambers contained a small vent (10 cm coiled 1/8" diameter copper pipe) to ensure a constant pressure during measurements. The opaque chamber fluxes of $CO_2$ provided an independent estimation of ER. This helped characterize ER given the challenges with standard NEE partitioning techniques at high latitude sites during the Arctic summer as noted in section 2.5.1.

Chamber flux measurements were made between 9:00 and 17:00 starting at a different collar set each day to randomize the sampling order to avoid a bias due to diurnal patterns. During gas flux measurements, the chambers were sealed to the top of the collars within a groove filled with water and five 24 mL air samples were collected into evacuated 12 mL vials sealed with doubled septa. Each vial contained a small amount of magnesium perchlorate to dry the air sample. Samples were collected at 0, 5, 10, 15 and 20 minutes after the chambers were set on the collars. Air within

the chamber was mixed with a 60 mL syringe attached to a three-way stopcock before each air sample was taken. Samples were stored until analysis one month later at Carleton University. The integrity of the vials through shipping, storage and analysis was confirmed using a subset filled with helium before the field season began.

Concentrations of $CO_2$, $CH_4$ and $N_2O$ were determined using a CP 3800 gas chromatograph (Varian Inc., Pao Alto, CA, USA) as described by Wilson and Humphreys (2010). Three replicates of five $CO_2$/ $CH_4$ standards varying from

383.1 to 15212.6 ppm $CO_2$ and from 1.08 to 22.11 ppm $CH_4$ were included in every set of measurements to create a linear relationship between gas concentration and chromatogram area. The chamber fluxes of $CO_2$ and $CH_4$ ($F_C$) were calculated as follows:

$$F_C = \frac{VP}{ART}\frac{dc}{dt} \tag{2}$$

where ($dc/dt$) is the linear rate of change in the mixing ratio of the gas, $A$ is the chamber area (0.0464 m$^3$), $V$ is the

chamber volume (between 0.0182 and 0.0242 m$^3$ adjusted for collar depth at each collar location), $R$ is the ideal gas constant, $P$ is pressure in $Pa$ and $T$ is the air temperature in Kelvin. $P$ and $T$ values corresponding to the time of each measurement were obtained from the EC station. Visual inspection of the linear trend of gas concentrations (dc/dt) was used to identify and remove spurious point measurements associated with analysis errors, leaking chambers (isolated decreases in concentration) and contamination or ebullition events (isolated increases in concentration)

(0.3%, 0.7%, and 2.0% of CO2 samples and 2.1%, 0.5%, and 1.1% of CH4 samples, respectively). In all flux measurements, at least three or more gas samples remained so that dc/dt and its coefficient of determination (R2) were determined using least squares linear regression. We did not use R2 as an additional quality control criterion as many of our CH4 fluxes were near zero and tended to have low R2 values due to only small variations in the point sample concentrations (see also Clark et al., 2020). 40% and 32% of the 227 CH4 flux measurements and 97% and 92% of

the 227 CO2 flux measurements had R2 over 0.80 and 0.90, respectively. No flux measurements were removed from

the analysis. Positive fluxes indicate emissions of gases to the atmosphere and negative fluxes indicate uptake by the surface.

### 2.4.1 Upscaling

Chamber fluxes of ER were upscaled from the plot scale (individual chamber) to the footprint scale using the footprint weighted average method and to the basin scale using the area weighted average method (Budishchev et al., 2014). The chamber ER and air temperature from the EC tripod ($T_a$) were used to determine $R_{10}$, the base respiration at 10 C°, and $Q_{10}$, the temperature sensitivity coefficient, using eq 3 for five of the six surface classes (Fig 1) (Laforce, 2018) (Table 3).

$$\text{ER} = R_{10}Q_{10}^{\frac{(T_a-10)}{10}}$$ (3)

Half hourly footprint scale estimates ($\text{ER}_{FS}$) were calculated by multiplying ER derived from eq. 3 for each surface class by the footprint source area fraction and summing over classes. Basin scale estimates ($\text{ER}_{BS}$) were estimated the same way but using the mean source area fractions of the basin (Table 2). As there were no open water class ER estimates, ER from open water was assumed to be zero.

In contrast to ER, there are no standard empirical functions to estimate temporal variations in NME. Instead, we used ordinary least squares regression (OLS) to estimate NME. The most important environmental controls over $F_{CH4}$ were *VWC* and $T_s$ (discussed below). Continuous observations of these factors at the flux chambers were not available, instead chamber NME were grouped by vegetation class and fit to *VWC* and $T_s$ measured in the soil pits near the EC station. Half hourly footprint scale ($\text{NME}_{FS}$) and basin scale ($\text{NME}_{BS}$) estimates were then estimated using the OLS parameters for each surface class using the same procedures for $\text{ER}_{FS}$ and $\text{ER}_{BS}$.

### 2.5 Factor Selection and Gap Filling

We used an exploratory approach to identify the smallest set of factors that best predicted half hourly EC-derived NEE and NME without overfitting the dataset using a series of neural networks (NN). We started with 10 factors: four meteorological variables [ (*PPFD*), $T_a$, vapor pressure deficit (*VPD*) computed using the $T_a$ and Rh data, three-dimensional wind speed (*U*) measured using the CSAT3 sonic anemometer], two soil variables [(*VWC*) and $T_s$ averaged between the two soil pits near the EC tripod], and four source area fractions [Shrub ($F_{Shrub}$), Grass ($F_{Shrub}$), Sedge ($F_{Sedge}$), and Upland, ($F_{Upland}$)]. The four source area variables correspond to surface classes sampled by the chambers. We excluded Water ($F_{Water}$) and Sparse ($F_{Sparse}$) fractions because its average contribution to the EC observations was only 0.2% and 2.2%, respectively, and there were no chamber measurements for the Water class while chamber measurements indicated ER was low and NME was not significantly different from zero for the Sparse class. A number of these prediction factors were highly correlated but it was necessary to include them so the model could account for source area heterogeneity.

The NNs were trained iteratively on bootstrapped datasets. First NN were trained on each factor individually and the one with the lowest MSE was selected. Next, NN were trained on that factor in combination with one of the remaining nine. The best performing additional factor was again selected and this process was repeated until MSE failed to

improve. The most parsimonious model was identified using the one standard error (*SE*) rule. Dybowski and Roberts

(2001) give the standard error of a bootstrap estimate of a given error metric (e.g., $\theta = MSE$) to be

$$SE_{boot}(\theta) = \sqrt{\frac{1}{B-1}\sum_{b=1}^{B}(\theta_b - \theta_{boot})^2} \tag{4}$$

where $\theta_{boot}$ is the mean of the bootstrapped samples. The smallest set of factors where $\theta_{boot}$ was within one $SE_{boot}$

of the minimum $\theta_{boot}$ for both NEE and NME were selected for further analysis. The outputs from the selected models

are referred to as $NEE_{NN}$ and $NME_{NN}$, respectively. NN modelling was done using the Keras Python library (Chollet

et al., 2015), see the Appendix A for a more detailed explanation of the NN analysis.

Multiple Imputation (MI) was then used to gap fill the NEE and NME with the $NEE_{NN}$ and $NME_{NN}$, respectively

(Vitale et al., 2018). Of the 1296 half hourly flux observations 28.9% of $F_{CO2}$ and 31.3% of $F_{CH4}$ were missing or

filtered out. There were a few gaps in the source area fractions needed to gap-fill the flux time series because the

footprint function is not valid when $u_* < 0.1$ m s$^{-1}$. When source area fractions were missing, they were gap-filled by

using the mean source are fraction observed for winds within ± 5˚of the observed wind direction. The meteorological

and soil data were continuous and did not need to be gap-filled.

### 2.5.1 Flux Partitioning

NEE is negative when there is net uptake of $CO_2$ by the ecosystem and positive when there is net emission. ER and

GPP are always positive, ER represents the sum of heterotrophic and autotrophic respiration and GPP represents

photosynthetic uptake of $CO_2$. Night-time NEE observations (e.g., *PPFD* <= 10 µmol m$^{-2}$ s$^{-1}$) are typically used to

quantify ER because GPP ~ 0 (Aubinet et al., 2012). We fit the limited night-time EC observations available (n=95)

to equation 3 for comparison with the ER measured using the chambers. We used the fitted values to model daytime

ER and approximate NEE by fitting the daytime data to a light response curve Aubinet et al. (2012).

$$NEE = \frac{1}{2c}\left(\alpha PPFD + \beta - \sqrt{(\alpha PPFD + \beta)^2 - 4\alpha\beta c PPFD}\right) + ER \tag{5}$$

where $\alpha$ is the initial slope of the light response curve, $\beta$ is GPP at saturation, and $c$ is a curvature parameter. These

estimates are referred to as $ER_{Q10}$ and $NEE_{Q10}$.

Some NN analyses of NEE have trained separate models for night-time and daytime conditions for partitioning

purposes (Papale & Valentini, 2003). However, these methods are not practical during the Arctic summer as the sun

did not set at Illisarvik until July 28[th], over halfway through the study period. There were not enough night-time

samples to train a separate NN. Instead, we estimated ER by calculating $NEE_{NN}$ at PPFD = 0 µmol m$^{-2}$ s$^{-1}$ for all

observations, henceforth referred to as $ER_{NN}$. This is a projection outside of the observed parameter space resulting

in greater uncertainty and a wider confidence interval around $ER_{NN}$ than $NEE_{NN}$. Calculation of confidence intervals

for NN outputs is discussed in Appendix A.


### 2.5.2 Factor Analysis

The trained NNs were used to investigate how individual factors influenced NEE and NME. The partial first derivative

of the model response to one controlling factor was calculated while keeping all other inputs fixed. For example, the

partial first derivative, $\frac{\partial \text{NEE}}{\partial \text{PPFD}}$ , is an approximation of the NEE light response curve under a specific set of conditions.

Similarly, $\text{NME}_{NN}$ can be used to approximate NME response to controls like $VWC$ or $T_s$. For both fluxes, the selected models contained at least one source area fraction variable, indicating the vegetation type(s) which had significant influence over NEE and NME. Additionally, we mapped $\text{NEE}_{NN}$ and $\text{NME}_{NN}$ to 100% coverage for individual surface classes to see how fluxes at Illisarvik may change as vegetation succession continues. For example, to project to 100% Sedge coverage, we set the other surface classes to 0% and left the other environmental factors unchanged. This allows for an estimation of how carbon fluxes may change if vegetation succession leads Illisarvik to look more like the DTLBs studied in Alaska.

## 3 Results

During the 29-day study, half-hourly $T_a$ and $T_s$ ranged between 0.4 and 26.2°C and 4.4 and 11.0°C, respectively (Fig 2a). Day length and maximum solar altitude decreased from 24 hours to 19.25 hours and 41.6˚ to 35.4˚, but daily $PPFD$ was more influenced by variations in cloud cover. Precipitation (19 mm) fell on 14 of the 28 days with trace snowfall on three of those days, but $VWC$ of the soils decreased throughout the period (Fig 2b). At the onset of the study period, $VWC$ was high and soils were saturated with ponding in the sedge areas. By the end of the study most of this surface water had dried up. On July 11[th] average thaw depth (cm) was 37, 45, 51, 64, 81 at Upland, Sedge, Grass, Shrub, and Sparse classes, respectively. By August 6[th], average thaw depth had increased to 45, 62 and 66 cm at Upland, Sedge and Grass surface classes and over 100 cm at both the Shrub and Sparse classes.

A strong low-pressure system stalled off the coast between day of year (DOY) 199 and 204. This caused westerly winds to occur much more frequently than is typical for July and August. The 50%, 80% and 90% flux $F_{Clim}$ contours are shown in Fig 1a. Mean source area fractions indicate the EC observations were skewed towards the Grass surface class and under-sampled the Shrub class, but the range of surface classes sampled was diverse enough to allow for testing of the impact of source area fraction on the fluxes (Table 2).

### 3.1 EC Observations

Half hourly observations of $F_{CO2}$ and $F_{CH4}$ along with the $\text{NEE}_{NN}$ and $\text{NME}_{NN}$ used to gap-fill the time series are shown in (Fig 2c & d). Gap-filled daily NEE ranged from -3.7 to -0.2 g C-CO$_2$ m$^{-2}$ d$^{-1}$ with a mean -1.5 [CI$_{95\%}$ ± 0.2] g C-CO$_2$ m$^{-2}$ d$^{-1}$. Day to day variability was considerable but there was no notable trend in NEE over the peak growing season. The half hourly NEE during the study period reached a minimum of -10.4 µmol CO$_2$ m$^{-2}$ h$^{-1}$ just before solar noon and peaked at 4.7 µmol CO$_2$ m$^{-2}$ h$^{-1}$ around midnight (Fig 2c). $\text{NEE}_{NN}$ was used to gap-fill the flux data because it was in good agreement with $F_{CO2}$ observation ($r^2 = 0.91$). Daily $\text{ER}_{NN}$ was estimated to be 2.2 [CI$_{95\%}$ ± 0.9] g C-CO$_2$ m$^{-2}$ d$^{-1}$ with corresponding GPP of 3.7 g C-CO$_2$ m$^{-2}$ d$^{-1}$. $\text{ER}_{NN}$ was in poor agreement ($R^2 = 0.35$, n= 95) with night time $F_{CO2}$ observations. For comparison, Eq. 3 provided a better fit ($R^2 = 0.47$) with night-time EC data, and $\text{ER}_{Q10}$ was estimated to be 3.0 g C-CO$_2$ m$^{-2}$ d$^{-1}$. However, $\text{NEE}_{Q10}$ did not fit $\text{F}_{CO2}$ as well ($r^2 = 0.80$) as $\text{NEE}_{NN}$.

Gap-filled daily NME was modest and decreased over the study period. It ranged from 2.0 to 25.1 mg C-CH$_4$ m$^{-2}$ d$^{-1}$ with a mean of 8.7 [CI$_{95\%}$ ± 0.4] mg C-CH$_4$ m$^{-2}$ d$^{-1}$ (Fig 2d). $\text{NME}_{NN}$ was used to gap-fill the flux data because it

provided a reasonable fit ($r^2 = 0.62$) to $F_{CH4}$ observations. NME did not constitute a significant component of the carbon balance and thus the flux footprint area was a carbon sink during the peak growing season with negative GWP after accounting for the greater GWP of $CH_4$.

### 3.2 Chamber Observations

ER was highest in the Sedge, Upland, and Grass classes where fluxes were very similar at 5.5 [$CI_{95\%} \pm 1.2$], 5.4 [$CI_{95\%} \pm 1.2$] and 4.9 [$CI_{95\%} \pm 0.7$] g C-$CO_2$ m$^{-2}$ d$^{-1}$. Shrub ER was significantly less (3.5 [$CI_{95\%} \pm 0.6$] g C-$CO_2$ m$^{-2}$ d$^{-1}$) than the other vegetated classes and Sparse ER was the lowest among the classes (2.0 [$CI_{95\%} \pm 0.3$] g C-$CO_2$ m$^{-2}$ d$^{-1}$) (fig 3a). The $Q_{10}$ and $R_{10}$ values also differed between vegetation classes: ER in the Sedge was the most sensitive to changes in air temperature and modelled values provided the best fit ($R^2 = 0.82$) to observations. Upland and Grass had the highest base respiration and fit observations moderately well (Table 3).

NME was more variable between vegetation classes than ER (Fig 3b & c). Sedge was a very strong $CH_4$ source at 114.7 [$CI_{95\%} \pm 15.3$] mg C-$CH_4$ m$^{-2}$ d$^{-1}$. Shrub and Grass were very weak sources, 0.7 [$CI_{95\%} \pm 0.3$] and 0.4 [$CI_{95\%} \pm 0.3$] mg C-$CH_4$ m$^{-2}$ d$^{-1}$, respectively. Sparse was neutral. Upland was a net $CH_4$ sink -1.1 [$CI_{95\%} \pm 0.4$] mg C-$CH_4$ m$^{-2}$ d$^{-1}$. Sedge and Shrub were NME were positively correlated with $T_s$ (r=0.61, p < 0.01; r=0.35, p = 0.04) respectively and $VWC$ (r=0.58, p < 0.01; r=0.5, < 0.01) respectively. They also had a positive correlation with $T_a$, while Upland NME was negatively correlated with $T_a$. Grass and Sparse did not have any significant correlations.

Footprint-scaled chamber fluxes were 59% and 47% higher than $ER_{NN}$ or gap-filled NME, respectively. Mean $ER_{FS}$ was 3.5 g C-$CO_2$ m$^{-2}$ d$^{-1}$ [$CI_{95\%} \pm 0.1$], it fit $ER_{Q10}$ very well ($R^2 = 0.95$) as would be expected and $ER_{NN}$ moderately well ($R^2 = 0.46$). Mean $NME_{FS}$ was 12.8[$CI_{95\%} \pm 1.3$] mg C-$CH_4$ m$^{-2}$ d$^{-1}$, it did not fit $NME_{NN}$ well ($R^2 = 0.30$). At the basin scale, $ER_{BS}$ (3.4 [$CI_{95\%} \pm 0.1$] g C-$CO_2$ m$^{-2}$ d$^{-1}$) was slightly lower than $ER_{FS}$ because of the exclusion of upland areas. $NME_{BS}$ was higher (15.2 [$CI_{95\%} \pm 0.1$] g C-$CO_2$ m$^{-2}$ d$^{-1}$) because of the greater sedge fraction in the basin than the footprint because the (Table 2).

### 3.3 NEE Response to Environmental Factors and Vegetation Type

$NEE_{NN}$ ($r^2 = 0.91$) was estimated using four factors: $PPFD$, $VPD$, $VWC$, and $F_{Shrub}$. $PPFD$ is the primary control over NEE: a NN trained on $PPFD$ alone provided a reasonable fit ($r^2 = 0.83$). The three additional factors, $VPD$, $VWC$, and $F_{Shrub}$, helped $NEE_{NN}$ fit a wider variety of conditions. Examining the partial first derivative of $NEE_{NN}$ under different conditions provides interpretation of the modelled light response curves (Fig 4). The minimum values represent the peak light use efficiency and are analogous to $\alpha$ in eq. 5 (Fig 4b). With increasing $PPFD$, light use becomes less efficient and approaches zeros as the light response nears light saturation (Fig 4b).

$VPD$ was a secondary control over NEE. Increasing $VPD$ increased peak light use efficiency and net $CO_2$ uptake until a threshold, above which it had a strong limiting effect (Fig 4a & b). For example, under dry atmospheric conditions (e.g. VPD = 1.5 kPa), peak light use is less efficient (-12 nmol $CO_2$ $\mu$mol$^{-1}$ photon) than under more humid conditions (-18 nmol $CO_2$ $\mu$mol$^{-1}$ photon). The value of this VPD threshold was dependent upon soil moisture: from 1 kPa when $VWC$ was highest to 0.2 5Pa when $VWC$ was low. Mapping $NEE_{NN}$ and $ER_{NN}$ at $F_{Shrub} = 100\%$, $F_{Shrub} = 0\%$, and $F_{Shrub} = 36\%$ ($F_{Clim}$), shows that $VWC$ and $F_{Shrub}$ were the primary controls over ER and thus influenced NEE (Fig 4c & d).

We can see from the partial first derivates of $NEE_{NN}$ that increasing *VWC* increases ER from Shrub areas. In the absence of shrubs, increasing *VWC* inhibits ER although it is important to note that variations in VWC were subtle ranging from 51.7% to 59.0%. The partial first derivative of $NEE_{NN}$ shows that *VWC* slightly limits NEE from non-Shrub areas and significantly reduces it in Shrub areas.

**3.4 NME Response to Environmental Factors and Vegetation Type**

$NME_{NN}$ ($r^2 = 0.62$) was estimated using five factors: $F_{Sedge}$, $F_{Shrub}$, *VWC*, $T_S$, and *U*. NME was more variable and less dependent on any one factor than NEE which is why the $NME_{NN}$ needed an extra factor and had a lower $r^2$ score. Source area had a significant effect on NME, and it was encouraging that the model contained $F_{Sedge}$ and $F_{Shrub}$ since Sedge and Shrub were the strongest $CH_4$ source and largest footprint component, respectively. These two factors can combine to map NME under three general situations: we can extrapolate to $F_{Sedge} = 100$ % and $F_{Shrub} = 0$ % or $F_{Sedge} = 0$ % and $F_{Shrub}$ 100 %, or represent actual $F_{Clim}$ where $F_{Sedge} = 11$% and $F_{Shrub} = 37$% (Table 2). Some upland tundra was included in the $F_{Clim}$ estimate, which reduced NME.

*VWC* was the primary climatic driver identified by $NME_{NN}$. Wetter soils had a consistent positive effect on NME which was strongest when $F_{Sedge}$ was high (Fig 5a & b). Between driest and wettest conditions, estimated NME increased: by an order of magnitude at $F_{Sedge} = 100$ %, 4-fold at $F_{Shrub} = 100$%, and from neutral to a source at $F_{Clim}$ (Fig 5a). Higher $T_s$ generally had a negative effect on NME (Fig 5c & d). The negative correlation between $T_s$ and VWC ($r = 0.54$, $< 0.01$) may have contributed to this result. $NME_{NN}$ performance improved less with the addition of *U* indicating the $NME_{NN}$ was near saturation and its effects are less relevant. Higher U had a weak limiting effect on NME when *VWC* was high and increased NME when *VWC* was low (not shown).

**4 Discussion**

**4.1 Carbon Balance and Controlling Factors**

Compared to other DTLBs, Illisarvik has drier soils and greater shrub and grass cover (Table 4). Peak growing season $CO_2$ uptake at Illisarvik was greater than at most wet sedge-dominated DTLBs (Table 4; Zona et al. 2010, Sturtevant and Oechel, 2013; Lara et al. 2015). These differences may be due to differences in the periods of observation and year to year variability but may also be due to the presence of more productive shrubs and slightly warmer climate at Illisarvik. Mean 1980-2010 $T_a$ at Utqiagvik (formerly Barrow, AK) is -11.2 °C (US National Climate Data Centre, 2020). Tuktoyaktuk, the closest station to Illisarvik is 1.1° warmer. Shrub cover is expected to have a number of impacts on the microclimate and carbon cycle of Arctic tundra (eg. Myers-Smith et all, 2011). Typically, greater deciduous shrub cover is expected to increase GPP as a result of greater leaf area and photosynthetic potential compared to graminoid-dominated tundra (Sweet et al. 2015; Street et al., 2018). GPP was greater at Ilisarvik compared to the young wet-sedge dominated DTLBs in Alaska (Zona et al., 2010). It was more similar to Katyk which has significant dwarf shrub cover, predominately *Betula nana* and *Salix pulchra* (van der Molen et al. 2007). Differences in ER among tundra environments can be related to substrate availability, soil moisture and temperature and thaw depth, among other factors (Sturtevant and Oechel, 2013). The 'snow-shrub hypothesis' (Sturm et al. 2001)

describes the potential for greater snow trapping in shrub communities which insulates soils in winter, leads to increased decomposition and nutrient availability and promotes further shrub growth. At Illisarvik, snow blowing in off the Arctic Ocean results in large snow drifts within the basin where snow depth correlates with vegetation height

(Wilson et al., 2019). Wilson et al. (2019) concluded that the soils within the Illisarvik basin were warmer than those of the surrounding dwarf-shrub tundra in part through these snow-shrub interactions. Although our chamber observations suggested Shrub ER is lower than ER from other vegetation classes, this may have been an artifact as the taller shrubs (>40 cm) could not fit inside the chambers. In another study, chamber ER increased with greater shrub cover in upland tundra (Ge et al., 2017). ER at Illisarvik was greater than the ER observed at both the young

wet-sedge DTLB in Barrow (Zona et al., 2010) and at the shrub/wet sedge DTLB at Katyk where thaw depth was much shallower (45 to >100 cm at Illisarvik vs. 25 to 40 cm at Katyk; van der Molen et al. 2007). The importance of $F_{Shrub}$ in describing temporal variations in half hourly NEE within the flux footprint at Ilisarvik is further evidence of the importance of shrub cover on tundra carbon cycle processes in this environment.

$PPFD$ and $VPD$ were the most important factors for predicting half hourly NEE. This was to be expected as they are

typically the primary controls over GPP (Aubinet et al., 2012). The limiting effects of VPD are consistent another study using NN to analyse NEE at a deciduous forest site (Moffat et al., 2012) and has been found at other tundra sites (Euskirchen et al. 2012; López-Blanco et al. 2017). $VWC$ was also important at Illisarvik. Zona et al. (2010) found $VWC$ could explain 70% of the variability in daily peak season ER in a young DTLB. Similarly, Kittler et al. (2016) found drier soils increased ER and decreased NEE after a wet tundra drainage experiment in Siberia, consistent with

our results at Illisarvik when $F_{Shrub}$ was low.

As expected, NME at Illisarvik was about half that observed at the Alaskan DTLBs sites where soils were wetter with greater sedge cover (Table 4, Zona et al., 2009; Lara et al. 2015). NME at Katyk was even higher than the Barrow DTLBs and had a significant impact on the greenhouse gas (GHG) balance for this site (van der Molen et al. 2007; Parmentier et al., 2011). In our NN modelling of NME at Illisarvik, $F_{Sedge}$ was the most important factor for predicting

half hourly $F_{CH4}$. Sedges are aquatic plant species with arenchymatous tissues that act as conduits for $CH_4$ from below the water table to the atmosphere and limits $CH_4$ oxidation by methanotrophs in aerobic surface soils (Lai et al. 2009). The inclusion of $F_{Shrub}$ further refined the model, allowing it to better fit the site-specific distribution of vegetation types. Budishchev et al. (2014) found shrub and sedge fraction had a significant influence on $F_{CH4}$ at Katyk. Vegetation type is the dominant control over NME across multiple tundra landscapes and our results further support

that (Davidson et al., 2016).

$VWC$ was the second most important factor, which was expected as $CH_4$ production occurs in anaerobic environments and has been linked to variability in $CH_4$ emission in many other studies (e.g. Zona et al., 2009; Nadeau et al., 2013; Olefeldt et al., 2013). Soil temperature ($T_s$) was the third most important factor. Higher $T_s$ increase the oxidation potential of methanotrophs (Liu et al., 2016; King and Adamsen, 1992), so this result was expected for the drier

portions of the basin and upland tundra. However, this was not expected for the sedge areas because most studies find NME in sedges is positively correlated to $T_s$ (Olefeldt et al., 2013). The negative correlation between $T_s$ and $VWC$ may partly explain this.

## 4.2 Upscaling

$ER_{FS}$ and $NME_{FS}$ were about 59% and 47% greater than the gap filled EC estimates. Discrepancies between EC and
chamber observations are common and have been attributed to differences in measurement techniques, the small sample size of chamber observations, and sampling bias since all chamber measurements were taken during the day with fair weather (Katayanagi et al., 2005; Chaichana et al., 2018). Meijide et al. (2011) found that chamber NEE could be up to twice as large as EC observations and Riederer et al. (2014) also found chamber NME estimates were about 30% higher than EC estimates. Others have been more successful, yielding upscaled chamber NME fluxes
within 10% of EC observations (Zhang et al., 2012; Budishchev et al., 2014; Davidson et al., 2017). A potential reason for the disagreement with $ER_{FS}$ may be the lack of direct observations by the EC system under low-light conditions. Another potential source of error for the upscaling is inaccuracies in the vegetation map.

## 4.3 Future Trajectories

Presently, peak growing season carbon uptake at Illisarvik is greater than similarly aged landscape features on the
Barrow Peninsula, Alaska and more similar to levels observed at Katyk, Siberia. NME is well below levels observed at any other DTLB studied, making this site a stronger GHG sink than other DTLBs. However, the basin at Illisarvik will continue to evolve and the trajectory it takes could significantly alter its carbon balance. Historically, DTLBs on Richards Island and the Tuktoyaktuk Peninsula evolve into sedge wetlands, as do DTLBs on the Barrow Peninsula (Ovendend, 1986; Lara et al., 2015). Active maintenance of the outlet channel at Illisarvik has artificially lowered
soil moisture and flooding and potentially limited this transition thus far (C. Burn, personal communication 2016).
If Illisarvik follows the same trajectory as older DTLBs in the area and becomes dominated by sedge wetlands, NME will increase significantly. Figure 5a shows that with extrapolations to full Sedge cover ($F_{Sedge}$ = 100%), NME would be similar to values on the Barrow Peninsula (Zona et al., 2009). If the basin instead transitions into a shrub dominated DTLB similar to those of Old Crow Flats, Yukon (Lantz et al., 2015), $NME_{NN}$ would remain similar to current levels
meaning the basin would remain a weak source of $CH_4$. These are projections well beyond $F_{Clim}$ fractions observed so confidence in the specific values predicted is low.
The effects of changing shrub/sedge cover on Illisarvik's growing season NEE are less straightforward than on NME. Partly because Shrub cover had less overall influence on $NEE_{NN}$. Figure 4c. shows the model suggests ER decreases and NEE increases with increasing shrub coverage when soils are slightly drier, but has the opposite effect under
wetter conditions. To our knowledge, only few winter season (e.g. Zona et al. 2016) and no year-round studies of DTLB NEE and NME have been published to help evaluate the factors influencing carbon losses through the non-growing season months. Further observation year-round is needed to better understand the implications of continued vegetation change on the carbon balance of DTLBs such as Illisarvik.

## 5 Conclusions

This study investigated NEE, GPP, ER and NME in the Illisarvik experimental DTLB using EC and chamber data. To our knowledge this is the first such study conducted in a DTLB outside of the Barrow Peninsula or Siberia.

Illisarvik is a carbon sink during the growing season with NME only having a small effect on the net carbon balance. Our flux observations were generally in agreement with other studies but show how shrub-dominated DTLBs such as Illisarvik and Katyk in Siberia differ from sedge-dominated DTLBs on the Barrow Peninsula. Illisarvik's growing season net carbon uptake was greater than young and ancient DTLBs on the Barrow Peninsula and more similar to the shrub dominated ancient DTLB in Siberia. NME at Illisarvik was lower than all published DTLB studies likely due to better drainage and more diverse vegetation. A longer, more comprehensive study would be needed to resolve the annual carbon budget for Illisarvik.

Chamber measurements of ER and NME from different land cover classes within and outside the Illisarvik basin added context to the EC observations. Vegetation class (and associated difference in terrain and soil properties) had only a small but significant impact on NEE and ER but was one of the dominant controls over NME. Sedge areas were a strong source of $CH_4$, other vegetation types in the basin were weak sources, and upland areas were a net sink. These results suggest that NME in particular will change as the Illisarvik DTLB vegetation communities continue to evolve.

**Appendix A: Neural Networks analysis and uncertainty calculations**

Typically, NEE is gap-filled using flux-partitioning algorithms that model ER and GPP separately using $T_S$ and $PPFD$, respectively (e.g. Lee et al., 2017; Aubinet, 2012). However, this method requires night-time observations and thus does not perform well for Arctic summertime measurements due to the limited number of samples available during low light conditions. There are no widely agreed upon functional relationships for gap-filling NME since $CH_4$ production and consumption vary considerably both between different landcover types and environmental conditions. Some methods that have been used include: general linear models (GLM) (Zona et al., 2009), mean diurnal variation (Nadeau et al., 2014), and classification and regression trees (CART) (Nadeau et al., 2013; Sachs et al., 2008). We attempted to use a GLM and CART but they were not flexible enough to account for source area variability.

Neural networks (NN) are flexible machine learning methods that are ideally suited to perform non-linear, multivariate regression. They make no a priori assumptions about the functional relationships between the factors and responses. (Melesse and Hanley, 2005; Desai et al., 2008). NN are universal approximators; given enough hidden nodes a NN is capable of mapping any continuous function to an arbitrary degree of accuracy (Hornik et al., 1991). If all relevant climate and ecosystem information are available to a network, the remaining variability can be attributed to noise in the measurement (Moffat et al., 2010).

NN have been shown to be among the best performing methods for gap-filling NEE data for temperate forest and wetland sites (Papale et al., 2003; Moffat et al., 2007; Knox et al., 2016). They have also been used to gap-fill NME time series in sub-arctic wetlands, tundra sites, and wet sedge tundra (Dengel et al., 2013). NN have been used to identify and model factors influencing NEE and to partition NEE into ER and GPP (Moffat et al., 2010). NNs have even been used to upscale fluxes from the ecosystem level to the continental scale (Dou and Yang, 2018; Papale et al, 2003).

A NN approximates a true regression function $F(X)$:

$$F(X) = t(X) - \varepsilon(X) \tag{A1}$$

where $t(X)$ is the target function and $\varepsilon(X)$ the noise (Khosravi and Nahavandi, 2010). $X = [x_0, x_1, \dots, x_M]$ where $x_0 = 1$ is a bias term and $[x_1, \dots, x_M]$ are the independent variables. $M$ denotes the number of independent variables.

The network approximates $F(X)$ as $f(X, w)$ by mapping the relationship between $X$ and the target. Here we used feed-forward dense NN with a single hidden layer:

$$f(X, w) = \sum_{h=1}^{H} \beta_h g\left(\sum_{m=0}^{M} \gamma_{hm} x_m\right) \tag{A2}$$

$g(\cdot)$ is a non-linear transfer function, here we used the rectified linear activation unit (ReLu) (Anders and Korn, 1999). $H$ denotes the number of hidden nodes in the network and must be assigned before training. Too many hidden nodes

and the NN will overfit the training data, too few and it will underfit. Early stopping will prevent NN from overfitting training sets (Weigend, 1993; Sarle, 1995; Tetko et al., 1995). Therefore, it is more important to ensure a NN has enough hidden nodes to adequately map the target function (Smith, 1994). We set $H$ to a function M, the number of training samples (N), and the number of targets (1):

$$H = \frac{N}{a*(M+1)} \tag{A3}$$

This rule of thumb ensures a NN has sufficient flexibility to approximate the target response. The weights $w = [\beta_1 \dots \beta_N, \gamma_{10} \dots \gamma_{NM}]$ are randomly initialized and after each model iteration is updated by backpropagating the error through the network. $N$ denotes the number of observations or targets. The error metric most commonly used is the mean squared error, MSE:

$$MSE = \sum_{i=1}^{N}(f(X_i) - t_i)^2 \tag{A4}$$

The weights are adjusted in the direction that will decrease the error and training continues until a stopping criterion is reached. We chose to set aside 20% of the training data as a test set to be used for early stopping, and terminated training when the MSE of the test set failed to improve for 10 consecutive iterations.

Bootstrapping is used to account for model variability and estimate confidence and prediction intervals by training NN on $B$ different realizations of the dataset, where $B$ is the number of bootstrapped samples, we used $B = 30$ (Heskes,

1997; Khosravi & Nahavandi, 2010). An individual NN generates point outputs approximating a target function with no information on the confidence in those estimates (Khosravi & Nahavandi, 2010). However, there are usually multiple $f(X, w)$ that approximate $F(X)$ because of the random weight initializations (Weigend & LeBaron, 1994). As such, there are two sources of error we are concerned with, the accuracy of our estimation of $F(X)$ and the accuracy of our estimates with respect to the target. A confidence interval describes the first (e.g. $F(X) - f(X, w)$) while a

prediction interval describes the latter (eg. $t(X) - f(X, w)$) (Heskes, 1997). By definition, a prediction interval contains the confidence interval because:

$$t(X) - f(X, w) = [F(X) - f(X, w)] + \varepsilon(X) \tag{A5}$$

For b = 1 … B, a random sample with replacement of size $p$ is drawn from the original dataset. Setting $p$ equal to the size of the original dataset yields a set of $B$ training sets each containing approximately 67% of the original dataset.

The 33% leftover from each bootstrap sampled can be used for model validation (Heskes, 1997). The average of our ensemble of networks can then serve as our approximation of F(X):

$$F(X) = \frac{1}{B}\sum_{b=1}^{B} f_b(X, W) \tag{A6}$$

The variance of the model outputs is:

$$\sigma^2(X) = \frac{1}{B-1}\sum_{b=1}^{B}\left(f_b(X,W) - F(X)\right)^2 \tag{A7}$$

A confidence interval (CI) for $F(X)$ can be calculated as $F(X) \pm t_{(1-\alpha,df)}\sigma(X)$, where *tscore* is the students t-score, $1-\alpha$ is the desired confidence level, and df are the degrees of freedom which are set to the number of bootstrapped samples B. NN performance can be seen to improve with the inclusion of more factors, until the model saturates and becomes over-parametrized (Fig A1).

Random forests (RF) are said to be among the best performing gap filling methods for NME (Kim et al., 2020). and
it has been claimed that aggregating many regression trees in a RF prevents overfitting (Breiman, 2001;). We did not find this to be the case. Following the methods outlined in Kim et al. (2020): a RF with 400 trees and no restrictions on tree size fit $F_{CH4}$ nearly perfectly ($R^2 = 0.98$). Without considerable limitations on tree size, the RF will just learn the dataset rather than the relationships present. It is our view that this tree is extremely overfit, as highlighted by the example in Figure A2. Further, RF do not allow for straightforward visualization functional relationships in a dataset.
Plotting $F_{CH4}$ against VWC, which is the dominant environmental control identified does not reveal a meaningful relationship like Figure 5 a & c. You can look at an individual decision tree within the RF, but those are difficult to interpret beyond the first few splits, and each tree will be different. Lastly, RF are incapable of projecting beyond the parameter space observed which limited their applicability for this study (Fig A2). This presents an issue because may gaps in EC data arise from data filtering (e.g. clear calm nights, precipitation events) and are by definition outside
the parameter space observed.

**Data & Code availability**

Our data and code are available on github: https://github.com/June-Spaceboots/Illisarvik_CFluxes

**Author Contribution**

JS, AC, and GH designed the EC study. AL and EH designed the chamber study. JS collected, processed, and
analysed the EC data. AL and EH collected the chamber data with help from JS. AL and EH processed the chamber data. JS designed and conducted the NN analysis. JS prepared the manuscript with input from all co-authors.

**Competing interests**

The authors declare they have no competing interests.

**Acknowledgements**

We would like to thank: the staff at the Aurora Research Institute in Inuvik for providing logistical support, Chris Burn for allowing us to work at Illisarvik and the knowledge he shared, Alice Wilson for sharing vegetation survey data, and Tony Lewkowicz for collecting and sharing the drone images, and Rick Ketler for providing logistical

support. Funding for this study was provided by the Canada Foundation for Innovation, the Natural Sciences and Engineering Research Council of Canada, and the Polar Continental Shelf Program, Natural Resources Canada.

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

**Figures & Tables**

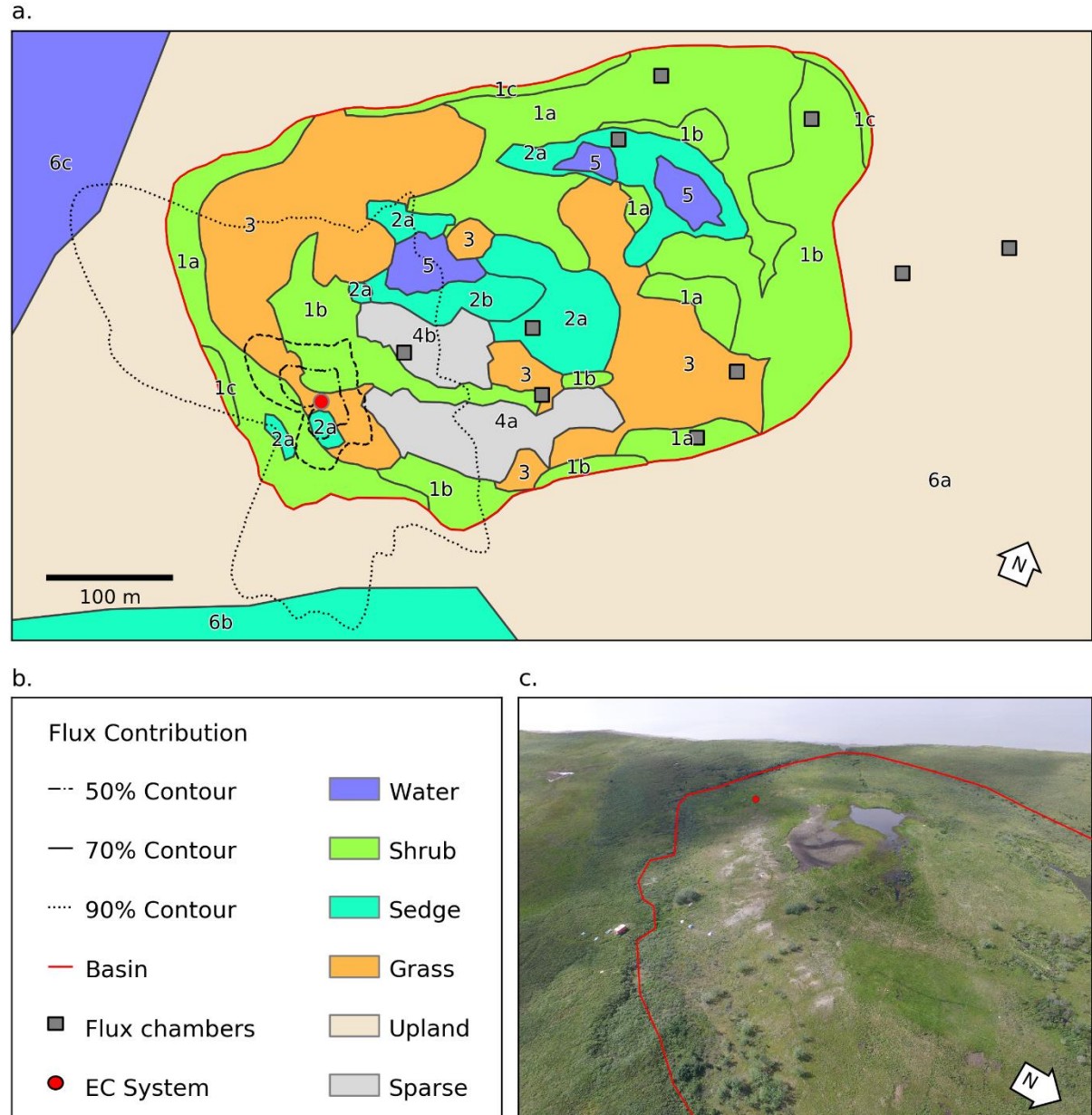

**Figure 1: a) Map of the distribution of vegetation classes at Illisarvik, with the footprint climatology ($F_{Clim}$) over the study period, the locations of the chambers and the eddy covariance (EC) system. The alphanumeric labels correspond to the unit codes in Table 1. b) Legend for the map in a. c) Oblique drone image of Illisarvik, take at 16:40 July 23$^{rd}$ 2016 view from E of DTLB towards W. The Basin and EC system are shown on the image using the same symbology as a).**


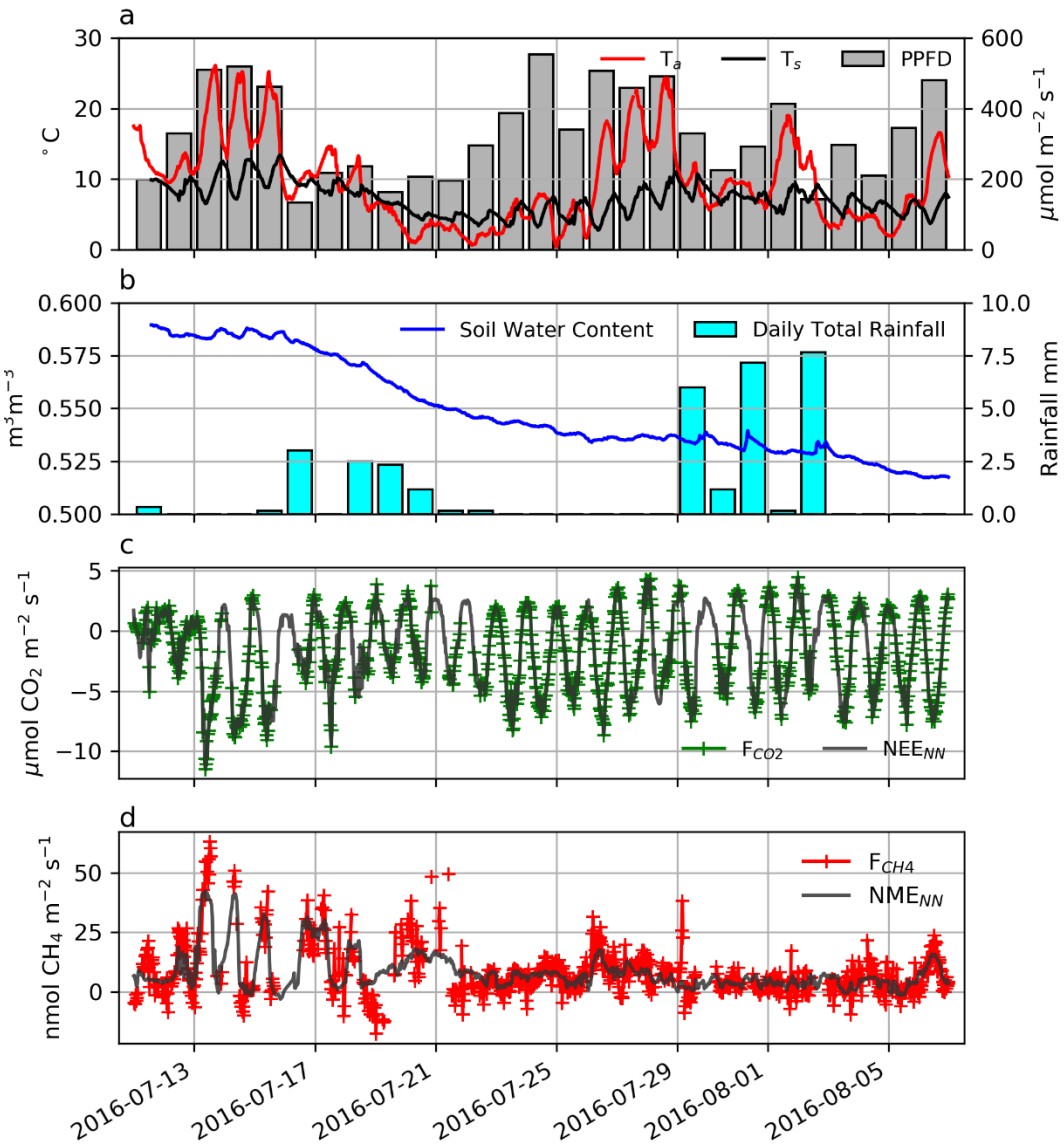

**Figure 2: a) Half hourly air and soil temperatures, displayed along with photosynthetic photon flux density (PPFD). b) Hourly soil volumetric water content and daily total precipitation. c) Half hourly $F_{CO2}$ (green) and $NEE_{NN}$ (grey), and d) half hourly $F_{CH4}$ (red) and $NME_{NN}$ (grey).**

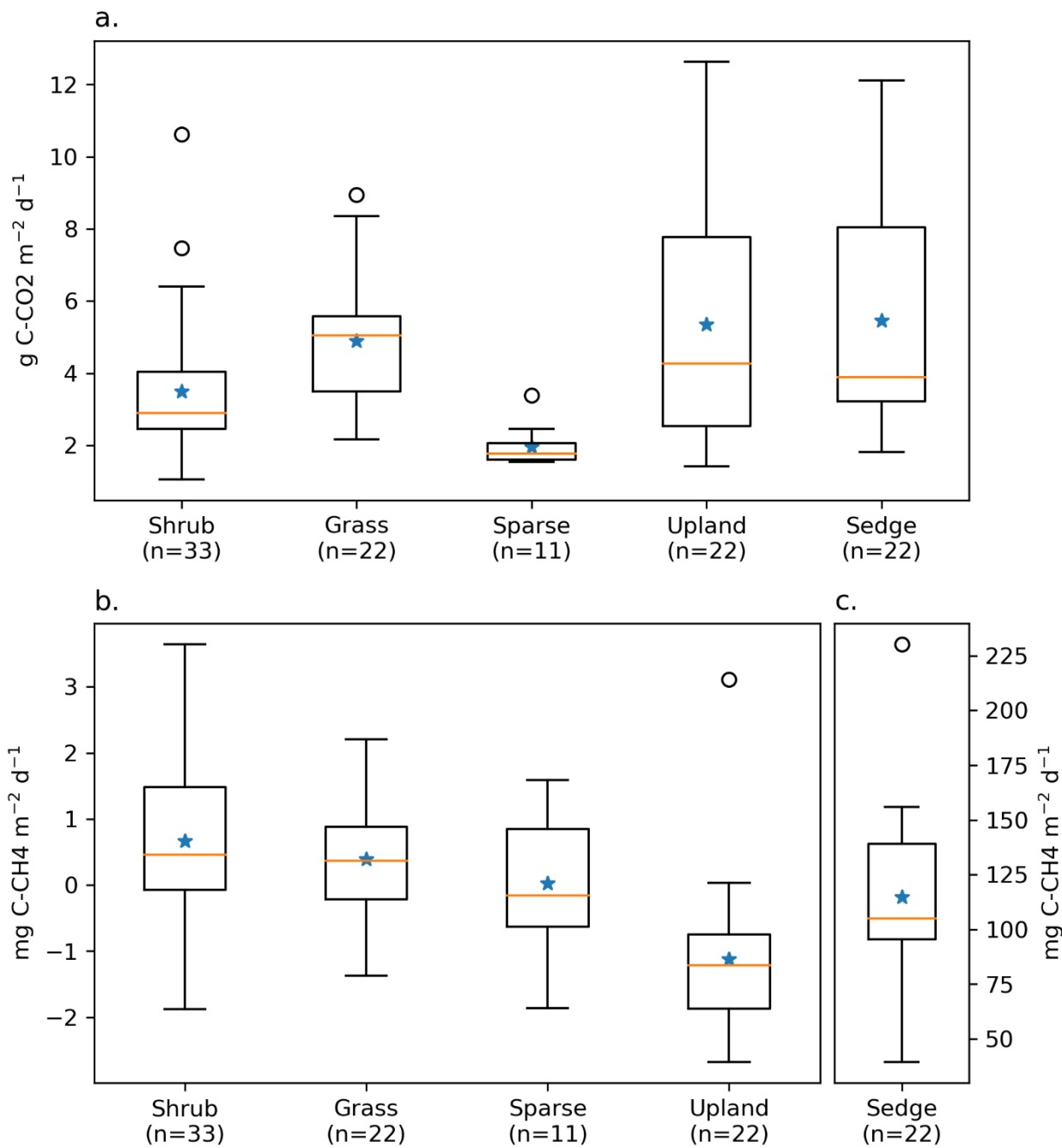

**Figure 3: Boxplot of a) ER, b) NME and c) NME fluxes measured using closed chambers, grouped by vegetation class. The orange lines represent the median, blue stars represent means, the boxes indicate the interquartile range (Q3-Q3), the whiskers indicate Q1 –(1.5*IQR) and Q3+(1.5*IQR), and the circles represent outliers extending beyond the whiskers. Note the scale for c) Sedge is different.**


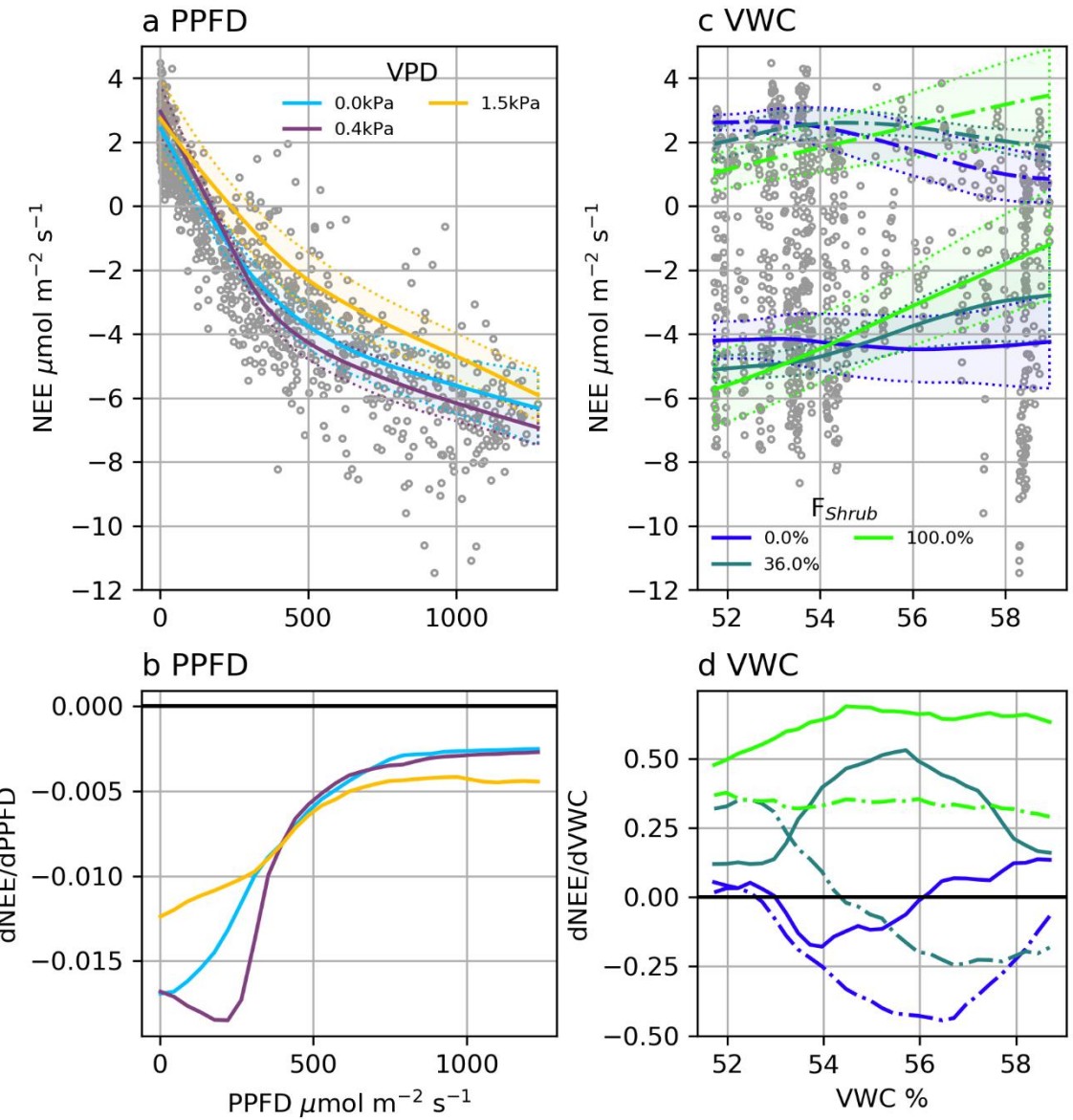

**Figure 4: a. Modelled NEE response to PPFD under different VPD conditions and b. the partial first derivatives of NEE with respect to PPFD. c. Modelled ER (dashed line) and NEE (solid line) response to VWC at different Shrub% and d. the partial first derivatives of ER (dashed lined) and NEE (solid line) with respect to VWC. NEE in c was calculated at PPFD = 600 μmol m² s⁻¹. The shaded areas in a & c are 95% confidence intervals and grey circles are the EC observations.**


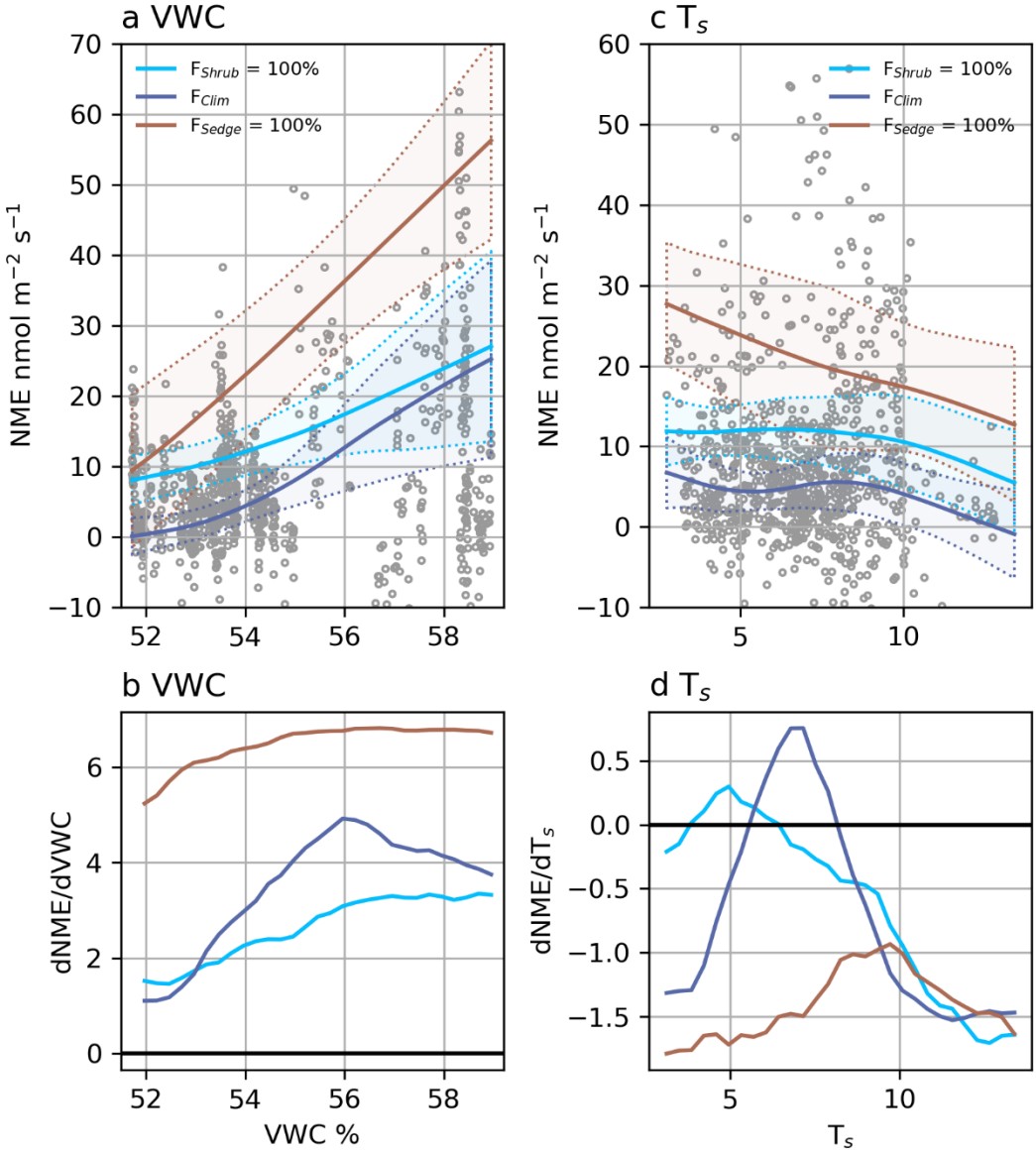

**Figure 5: a. Modelled NME response to VWC at different source area fractions and b. the partial first derivatives of NME with respect to VWC. c. Modelled NME response to $T_s$ at different source area fractions and d. the partial first derivatives of NME with respect to $T_s$. The shaded areas in a & c are 95% confidence intervals and grey circles are the EC observations.**

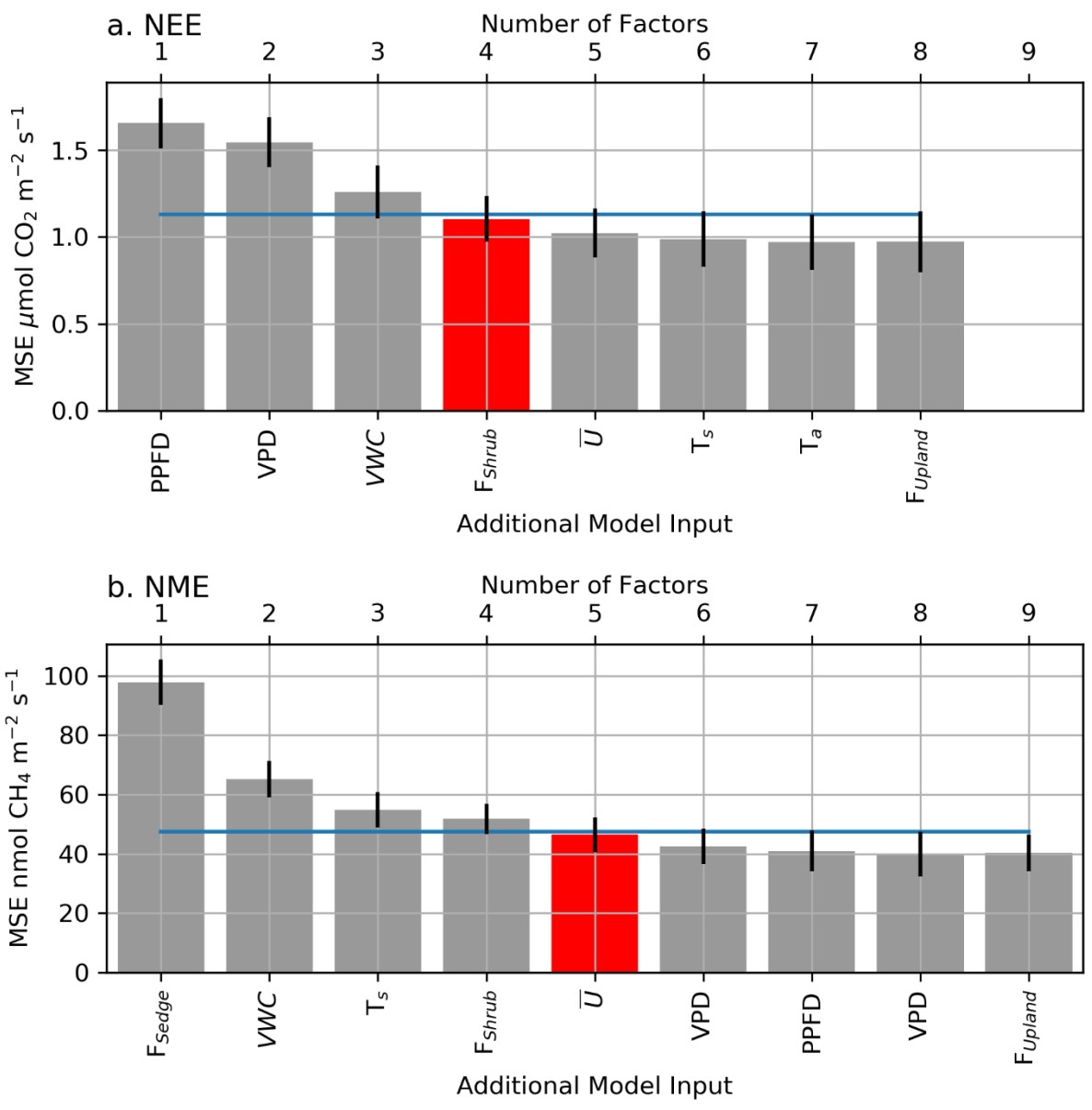


**Figure A2: The averaged mean squared error (θ) of the bootstrapped Neural Network model validation datasets, with error bars showing one standard error (SE). The x axis shows models of increasing size from left to right (1-9 factors), and the label indicates the factor added to the model at each step. The blue line indicates the 1-SE rule threshold and the red bar indicates the model selected by the 1-SE rule.**

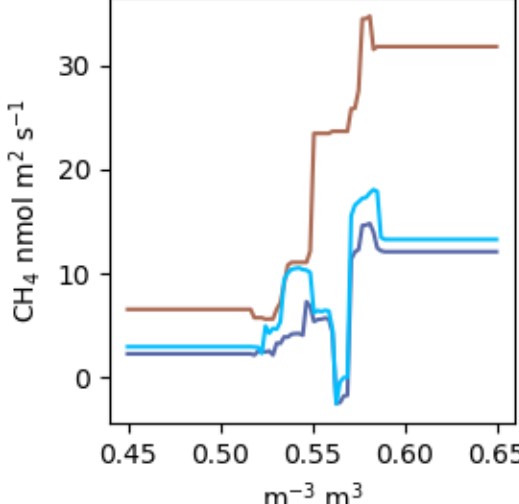


**Figure A2: F$_{CH4}$ estimated by a RF using the same factors as the NN model. The colours correspond to the scenarios in Fig 5a. *VWC* was estimated over the range 0.45 to 0.65.**





**Table 1: Dominant species or landscape feature within the vegetation/cover classes. Unit codes correspond to the map**
**Figure 1a.**

| Unit Code | Vegetation Class | Dominant Species/Landscape feature |
|---|---|---|
| 1a | Shrub | *Salix alaxnesis* (Tall Willow) |
| 1b | Shrub | *Salix glauca* (Low Willow) |
| 1c | Shrub | *Alnus viridis* subsp. *crispa* (Alder) |
| 2a | Sedge Marsh | *Carex aquatilis* (Sedge) |
| 2b | Sedge Marsh | *Arctophila fulva* (Pendant Grass) |
| 3 | Grass Meadow | *Pocacea* spp. (Grasses), *Eriophorum angustifolium* (Cotton Grass) |
| 4a | Sparse Cover | Sparse Vegetation |
| 4b | Sparse Cover | Bare Ground |
| 5 | Ponds | *Hippuris vulgaris* (Mare's Tail), Open Water |
| 6a | Outside of Basin | Dwarf shrub tundra: *Salix* spp. & *Betula nana* (Birch) |
| 6b | Outside of Basin | Fen |
| 6c | Outside of Basin | Ocean |

**Table 2: The surface cover class fractions of the basin, along with the mean source area fractions of the footprint climatology ($F_{Clim}$) and the range of source area fractions for individual half hourly observations shown in brackets.**

| Surface Class | Basin | $F_{Clim}$ |
|---|---|---|
| Shrub | 48.3 % | 36.0 % [0.0 – 79.0%] |
| Grass | 27.9 % | 39.0 % [1.1-78.1%] |
| Sedge | 12.3 % | 10.9 % [0.0 – 55.6%] |
| Sparse | 8.4 % | 2.2% [0.0 – 33.6%] |
| Water | 3.1 % | 0.2% [0.0 – 4.4%] |
| Upland | 0% | 6.2% [0.6 – 15.0%] |
| Outside Basin | 0% | 12.3% [0.2 - 28.0%] |

**Table 3: The ER temperature sensitivity ($Q_{10}$) and base respiration ($R_{10}$) estimated by Laforce (2018) and estimated from nighttime EC footprint observations.**

| | $Q_{10}$ | $R_{10}$ µmol m$^{-2}$ s$^{-1}$ | $R^2$ |
|---|---|---|---|
| Sedge | 2.1 | 3.8 | 0.82 |
| Upland | 1.9 | 4.1 | 0.55 |
| Grass | 1.6 | 4.0 | 0.55 |
| Shrub | 1.8 | 2.7 | 0.46 |
| Sparse | 1.0 | 1.9 | 0.01 |
| Night-time EC observations (n=100) | 1.6 | 2.9 | 0.47 |

**Table 4: Growing season (gs) daily range in eddy covariance-derived NEE and NME from drained thermokarst lake basins (DTLB) and other select wetland/coastal tundra sites across the Arctic. The period of studies measurements for the studies**

**observations are: a) mid-June – end of July b) June 12 – August 28, 2007, Fig 4 c) June 11 – August 25, 2011 d) upscaled chamber estimates, exact dates not specified, e) mean June 15 –August 31 2003-2006, f) July 5 – Aug 4, 2009.**

| Site | Site Characteristics | NEE<br>g C-$CO_2$ m$^{-2}$ d$^{-1}$ | NME<br>mg C-$CH_4$ m$^{-2}$ d$^{-1}$ | Studies |
|---|---|---|---|---|
| Illisarvik | Young DTLB, Low & Tall Shrub/Grass/Wet Sedge | -1.5 | 8.7 | (this study) |
| Various DTLB, Barrow Peninsula, Alaska | Young DTLB, Wet Sedge Tundra | -1.1[b], -0.9[d], -0.8[c] | 18.4[a], 26.1[d], 44.0[c] | Zona et al. 2009[a] & 2010[b], Sturtevant and Oechel, 2013[c]; Lara et al. 2015[d] |
| | Medium DTLB, Wet Sedge Tundra | -0.7[b], -0.6[d], -0.4[c] | 27.0[d], 41.3[c] | |
| | Old DTLB, Wet Sedge Tundra | -1.0[b], -0.4[d], 0.1[c] | 24.2[d], 38.7[c] | |
| | Ancient DTLB, Wet Sedge Tundra | 0.4[d] | 21.7[d] | |
| Katyky, Indigirka lowlands, Siberia | Ancient DTLB, Dwarf-Shrub and Wet Sedge Tundra | -1.3[e] | 36.0[f] | Van der Molen et al. 2007[e], Budishchev et al. 2014[f] |

