# Peer review of "Vegetation Influence and Environmental Controls on Greenhouse Gas Fluxes from a Drained Thermokarst Lake in the Western Canadian Arctic"

_Biogeosciences, 2019_

## Referee Comment (RC1) · Anonymous Referee #1 · 23 Mar 2020

This paper by Skeeter et al. looked at vegetation and environmental conditions influencing greenhouse gas exchange in a drained lake basin in the Western Canadian Arctic. I enjoyed the opportunity to review this paper and I thank the authors for what is a well written paper overall (but with some tweaks needed). Given the lack of studies outside of the Barrow Peninsula, is a worthy addition to the literature.

I will add however, I apologise, I am not an expert on the eddy covariance data cleaning, gap-filling and analyses and I am therefore unable to comment fully on those sections.

Introduction: I find this section well written if rather short. I think a little more context could be given for the reader. You could include more information from non-DLB work but still relevant arctic tundra literature.

Line 54: NEE should be ER – GPP.

Line 85: Can you include somewhere the dominant species found in each vegetation class. It would be interesting and useful to know what sort of sedge dominated the sedge class – is it *Carex aquatilis* or *Eriophorum angustifolium* for example?

Line 100: Completely an aside but COOL!

Section 2.3: I would remove any mention of N2O – you don't present the data, so it is unnecessary.

Line 150: Can you be more explicit with how many collars were used? 2 per site – 10 sites total? I know the main focus of this paper is not the collars, but I'm not sure if a replicate of 2 per vegetation type over an 11-day period is very representative.

Line 152: How soon after installation were the collars fluxed for the first time?

Line 154: Why not use a clear chamber so you could get GPP then cover with a dark sleeve in order to get ER?

Section 2.4 (and subsequently Appendix A): Unfortunately, I do not have the expertise in these methods so I do not feel comfortable commenting on it in a reviewer context.

Section 3: I think it would be better to separate this section out into Results and Discussion rather than combine them. As it stands, it's quite hard to follow.

Line 241: You only mentioned thaw depth twice? Why not measure it on each day chambers were used?

Section 3.1: I think this section needs an overhaul unfortunately. Many of the sentences do not make sense in their current format. For example: Lines 254-257: '*NEE was greater than (ie. less*

*carbon uptake) 255 EC observations of from four wetter, sedge dominated DTLB, where peak season NEE was -2.5 g C-CO2 m-2 d-1, ER (1.5 g C-CO2 m-2 d-1) was lower than at Illisarvik while GPP (4.0 g C-CO2 m-2 d-1) was slightly higher (Zona et al., 2010).'*

I also think it might be useful to separate the EC results and the chamber results into subsections. By referring your measured values to other studies in the results, it makes it hard to follow for the reader.

Further, by combining the results, there is a lack of discussing the results (for example, it feels like only section 3.4 is really doing this). It sadly reads as a lot of results statements and then suddenly we are at the conclusion.

Line 273: Why compare methane to ER here?

Line 286: Although discrepancies do occur between upscaling chamber measurements and EC measurements – some studies have done it successfully and I think it would be good to include here as a caveat;

- Budischev et al. 2014: Evaluation of a plot-scale methane emission model using eddy covariance observations and footprint modelling. Biogeosciences 11. 4651-4664

- Zhang et al. 2012: Upscaling methane fluxes from closed chambers to eddy covariance based on a permafrost integrated model. Global Change Biology, 18, 1428-1440.

- Davidson et al. 2017. Upscaling CH4 fluxes using high-resolution imagery in Arctic Tundra Ecosystems. Remote Sensing, 9, 1227; doi:10.3390/rs9121227

- Sachs et al. 2010 Environmental controls on CH4 emissions from polygonal tundra on the microsite scale in the Lena river delta, Siberia. Global Change Biology, 16, 3096-3110

I think more discussion of the results in the context of other GHG literature from other tundra ecosites would be useful. Although this study is focused on drained lake basins, the results are comparable to wet-sedge dominated tundra landscapes. I feel this would be a good addition and strengthen what is already a useful paper.

Figure 2a: Could you change the colour of the $T_a$ line? Red on orange is difficult to read.

Figure 3: I will leave this up to the author's discretion, but I wonder if this figure (and in fact, all figures) would benefit from having a plain white background. I find all the lines distracting. Especially when other lines are being used to annotate.

Figure 4c: Please use another three colours for Shrub. It is confusing that they are the same colour as VPD on the left-hand panel.

Figure 4 and 5: I think these figures would benefit from having a title for each panel – it was not clear to me initially the difference between Figure 5a and b. I think just putting VWC above left hand panel and $T_s$ above right hand panel, this would make it much clearer.

Tables: Caption should go above the Tables.

I think a table including the dominant vegetation species for each class would be super useful for the reader.

---

## Referee Comment (RC2) · Norbert Pirk (Referee) · 20 Apr 2020

General comments

The paper "Vegetation Influence and Environmental Controls on Greenhouse Gas Fluxes from a Drained Thermokarst Lake in the Western Canadian Arctic" by June Skeeter et al. reports $CO_2$ and $CH_4$ flux measurements from a permafrost tundra site in Western Canada. Eddy covariance and chamber flux measurements were taken during the growing season 2016, and analysed accounting for the spatial variability of vegetation cover. Statistical gap-filling and an analysis of the environmental controls of the fluxes is performed using artificial neural networks. I think the chosen methods are

properly applied and explained. The results are presented clearly and the conclusions are supported by the results. Also, the paper is very well written.

Given the remote and rather special site location, this study should be very valuable for the arctic carbon flux community. As the flux time series collected in your study may be used and referred to in future studies, it would be nice if you could present the time series in a more raw format than you do in Figure 2. For example, a plot of the 30-minute flux time series would help to understand the character of the data. This is also relevant, because I guess the performance and output of your NNs could be susceptible to noise or outliers in the EC time series. Also, several of your results (cf. Line 221 and Line 342) are based on extrapolations into parts of the parameter space where the flux response could be governed by processed not captured in your NNs. Perhaps these statistical uncertainties could be discussed.

I understand there is little research from DTLB sites, but it would be good in your discussion to relate your findings to those from other tundra sites with (and without) thermokarst. In this discussion, it would be good to elaborate further on the peculiarities of the artificial draining performed at your site. Given the title of this paper, readers will probably expect more of these aspects discussed.

Specific comments

Line 16: "During the study period". Please be more specific here, because the up-scaled average fluxes you mention in lines 18 and 20/21 don't tell much if you don't know the study period.

Line 24: Your abstract lacks a broader conclusion

Line 100: Could the grazing have a measurable effect on e.g. NEE? It could be a point to add to your discussion.

Line 116: You discarded a sector because its flow could be disturbed by the tower. But did you see this effect in any of your quality checks? Maybe it's not necessary to

discard this data.

Line 146: Maybe be more specific about the Python modules you used, otherwise this sentence adds very little to the understanding of your analysis.

Line 182: Shouldn't there be five times more vials than flux estimates, if you used 5 gas samples per flux measurement?

Line 292: "Random forest regression tree". Did you use only one decision tree, or the ensemble mean of several?

Line 296: Maybe refer to an equation defining alpha.

Technical corrections

Line 75: Did you really mean 100 m, or maybe km?

Lines 302/303: Pa, with a capital P

Line 310: "both"?

Please check and correct the names of your references in the text, as several have spelling mistakes ("Whalen and Reedburgh", "Merbould", "Meyer-Smith")

Figure 3b: Can you add a little bit of horizontal white space between the the Sedge plot and the rest? I think this could prevent confusion and make it clear that the y-axis for this box has a different scale

---

## Author Response (ED1)

**Response to Reviews: Vegetation Influence and Environmental Controls on Greenhouse Gas Fluxes from a Drained Thermokarst Lake in the Western Canadian Arctic**

The authors would like to thank the reviewers for their valuable comments and criticisms. Point by point responses to both reviewers are included in this document. Reviewer comments are in bold, responses are noted below each comment and shown in plain text. Changes to the manuscript are in red. Following that, is a marked-up version of the revised manuscript showing all changes made.

**Reviewer #1**

**This paper by Skeeter et al. looked at vegetation and environmental conditions influencing greenhouse gas exchange in a drained lake basin in the Western Canadian Arctic. I enjoyed the opportunity to review this paper and I thank the authors for what is a well written paper overall (but with some tweaks needed). Given the lack of studies outside of the Barrow Peninsula, is a worthy addition to the literature.**

**I will add however, I apologise, I am not an expert on the eddy covariance data cleaning, gapfilling and analyses and I am therefore unable to comment fully on those sections.**

**Introduction: I find this section well written if rather short. I think a little more context could be given for the reader. You could include more information from non-DLB work but still relevant arctic tundra literature.**
[Responses]
We added more context on why thermokarst landscapes are important (spatial extent, significant soil carbon storage) to the first paragraph [Lines 32-34].

Lake thermokarst landscapes are widespread in poorly drained, sedimentary permafrost lowlands with excess ground ice volume and constitute about a third of all thermokarst area (French, 2013; Olefeldt et al., 2016).

Further we added another paragraph on Arctic carbon budgets [Lines 40-49]. Including a review of pan-arctic NEE chamber studies by Virkkala et al. 2017.

Net ecosystem exchange (NEE), ecosystem respiration (ER) and gross primary productivity (GPP), where NEE=ER-GPP are lower in the Arctic than warmer regions but have significant seasonal cycles and variability between vegetation types (Virkkala et al., 2018). Future trajectories in NEE will in large part be governed by ER (Biasi et al., 2008; Cahoon et al.,2012). Dominant vegetation types in the Western Canadian Arctic are erect-shrub tundra and

wetlands (Walker et al., 2005).  Growing season NEE is typically negative across these units throughout the Arctic indicating a net CO2 sink as GPP exceeds ER in part due to cold and/or anoxic soil conditions (Virkkala et al., 2018; Lafleur et al., 2012).  Annual NEE can be positive or negative with large variation in GPP linked to annual weather variability (Virkkala et al., 2018, McGuire et al., 2009).  Arctic net methane exchange (NME) is positive because wetland areas are strong methane (CH4) sources while upland areas with better drainage can be net sinks (Whalen and Reeburgh, 1990; McGuire et al., 2009; Sturtevant and Oechel, 2013).

**Line 54: NEE should be ER – GPP.**

[Responses]

We corrected the equation to NEE = ER – GPP

**Line 85: Can you include somewhere the dominant species found in each vegetation class. It would be interesting and useful to know what sort of sedge dominated the sedge class – is it Carex aquatilis or Eriophorum angustifolium for example?**

[Responses]

We added a sentence to mention the specific species [Lines 90-92].

Current vegetation at Illisarvik is diverse relative to the dwarf-shrub tundra of the surrounding uplands (Table 1); the basin hosts a mix of woody shrubs (Salix spp., Betula spp., & Alnus spp), wetland vegetation (Carex aquatilis, Arctophila fulva, etc.), and various grasses (Pocacea spp.) (Wilson et al. 2019).

This information was partly available in Table 1.  We made the table more detailed to include the dominant species present within each class/subclass where known/applicable (see comment below).

*Line 100: Completely an aside but COOL!*

- Yes, it was an amazing thing to see.

**Section 2.3: I would remove any mention of N2O – you don't present the data, so it is unnecessary.**

[Responses]

We removed mentions of $N_2O$ throughout the manuscript.

**Line 150: Can you be more explicit with how many collars were used? 2 per site – 10 sites total? I know the main focus of this paper is not the collars, but I'm not sure if a replicate of 2 per vegetation type over an 11-day period is very representative.**

[Responses]

We updated the text to be more specific about numbers of replications per vegetation type [Lines 176-177].

There were three replicates (six collars) for the Shrub class, two for the Sedge, Grass, and Upland tundra, and no replicates for the Sparse class.

There were 19 collars and 10 sites. The bare ground site only had 1 collar, thus making 2 per sites for the rest. The number of collars that could be shipped in via helicopter were limited and there was a high amount of heterogeneity in soil and vegetation characteristics within the basin. The chamber study was designed to better understand the relationship between soil properties and carbon loss in a situation where permafrost had aggraded within the lake bed to potentially protect 'old' carbon from mineralization and accumulate 'new' carbon since the lake drained. We expected saturated soils (where there were wet sedges) to have higher organic carbon accumulation and be dominated by anaerobic respiration processes, which was interesting to us, therefore we chose two different wet sites populated by sedges. We also focused on different statures of will (low, tall and dense) as we expected different amounts of snow accumulation and different impacts on permafrost at these sites.

**Line 152: How soon after installation were the collars fluxed for the first time?**
[Responses]
Collars were installed on July 11[th] and first set of measurements was taken on July 12[th], so about 24h.

**Line 154: Why not use a clear chamber so you could get GPP then cover with a dark sleeve in order to get ER?**
[Responses]
We appreciate the comment. The chosen chambers were not designed for NEE measurements. Although measurements of GPP would have been informative, logistics limited us to use the existing collars, and the number of measurements we could make.

**Section 2.4 (and subsequently Appendix A): Unfortunately, I do not have the expertise in these methods so I do not feel comfortable commenting on it in a reviewer context.**

**Section 3: I think it would be better to separate this section out into Results and Discussion rather than combine them. As it stands, it's quite hard to follow.**
We have added a discussion (section 4) and rewrote the results to solely contain the objectively retrieved data, so we hope the manuscript to be more straightforward and easier to follow with separate "Results" and "Discussion" sections.

[revised manuscript text omitted]

**Line 241: You only mentioned thaw depth twice? Why not measure it on each day chambers were used?**

[Responses]

Thaw depth tends to increase over time but at different rates at different locations within the basin as a result of varying soil and surface properties. We measured thaw depth at the start and end of the measurement period to highlight these differences rather than develop a variable that could be related to the fluxes. In past studies we found that day to day variations in respiration correlate best with near surface soil temperature and moisture rather than thaw depth while spatial variations in average fluxes can sometimes correlate to max thaw depth. In the revised manuscript we refer to thaw depth more and use it to compare Illisarvik to Katyk Line 384

ER at Illisarvik was greater than the ER observed at both the young wet-sedge DTLB in Barrow (Zona et al., 2010) and at the shrub/wet sedge DTLB at Katyk where thaw depth was much shallower (45 to >100 cm at Illisarvik vs. 25 to 40 cm at Katyk; van der Molen et al. 2007).

**Section 3.1: I think this section needs an overhaul unfortunately. Many of the sentences do not make sense in their current format. For example: Lines 254-257: 'NEE was greater than (ie. Less carbon uptake) 255 EC observations of from four wetter, sedge dominated DTLB, where peak season NEE was -2.5 g C-CO2 m-2 d-1, ER (1.5 g C-CO2 m-2 d-1) was lower than at Illisarvik while GPP (4.0 g C-CO2 m-2 d-1) was slightly higher (Zona et al., 2010).'**

[Responses]

The entire section has been rewritten and the references to other work were moved out from the results to the discussion section. See previous comment response.

**I also think it might be useful to separate the EC results and the chamber results into subsections. By referring your measured values to other studies in the results, it makes it hard to follow for the reader.**

[Responses]

We agree and have separated the EC and chamber results into Sections 3.1 and 3.2. These changes are shown in the above comment about section 3.

**Further, by combining the results, there is a lack of discussing the results (for example, it feels like only section 3.4 is really doing this). It sadly reads as a lot of results statements and then suddenly we are at the conclusion.**

[Responses]

We added a separate "discussion" section (Section 4). These changes are shown in the above comment about section 3.

**Line 273: Why compare methane to ER here?**

[Responses]

We thought it is relevant to contextualize the differences. Methane emissions are far more spatially variable than ecosystem respiration. We revised the sentence to get rid of the comparison of the magnitude because that is less relevant, but left the point about spatial variability being enhanced [Line 316]. We think it is an important finding to show that NME is more influenced by spatial heterogeneity than ER.

NME was more variable between vegetation classes than ER (Fig 3b & c).

**Line 286: Although discrepancies do occur between upscaling chamber measurements and EC measurements – some studies have done it successfully and I think it would be good to include here as a caveat;**

- **Budischev et al. 2014: Evaluation of a plot-scale methane emission model using eddy covariance observations and footprint modelling. Biogeosciences 11. 4651-4664**
- **Zhang et al. 2012: Upscaling methane fluxes from closed chambers to eddy covariance based on a permafrost integrated model. Global Change Biology, 18, 1428-1440.**
- **Davidson et al. 2017. Upscaling CH4 fluxes using high-resolution imagery in Arctic Tundra Ecosystems. Remote Sensing, 9, 1227; doi:10.3390/rs9121227**
- **Sachs et al. 2010 Environmental controls on CH4 emissions from polygonal tundra on the microsite scale in the Lena river delta, Siberia. Global Change Biology, 16, 3096-3110**

[Responses]

We reviewed this literature and added it to the newly separated "discussion" section [Lines 417-418].

Others have been more successful, yielding upscaled chamber NME fluxes within 10% of EC observations (Zhang et al., 2012; Budishchev et al., 2014; Davidson et al., 2017).

We also decided to use the footprint weighted upscaling method discussed in Budishchev et al. (2014) for the chamber upscaling [Line 207-208], but it did not make an appreciable difference in the upscaled chamber ER or NME.

Chamber fluxes of ER were upscaled from the plot scale (individual chamber) to the footprint scale using the footprint weighted average method and to the basin scale using the area weighted average method (Budishchev et al., 2014).

**I think more discussion of the results in the context of other GHG literature from other tundra ecosites would be useful. Although this study is focused on drained lake basins, the results are comparable to wet-sedge dominated tundra landscapes. I feel this would be a good addition and strengthen what is already a useful paper.**

[Responses]

We added section 4.1 (see above comment) where we discuss NEE and NME observations at Illisarvik relative to natural shrub vs. sedge-dominated DTLB to highlight the differences among these environments rather than attempt to fully contrast Illisarvik to a myriad of arctic tundra types/sites. These comparisons are always challenging given different years, time periods within a year, instrumentation, and data presentation. However, we believe we make a strong argument that shrub vs. sedge-dominated DTLB have the potential to differ and Illisarvik differs in particular from all other DTLB in its low methane emissions. (Table 4). We now further highlight the important implications of vegetation succession on CO2 and CH4 fluxes at our site in the discussion section 4.3 "Future Trajectories". This is one of the key messages associated with DTLBs – they undergo relatively rapid vegetation change over a number of decades that will influence their C budgets.

Table 4: Growing season (gs) daily range in eddy covariance-derived NEE and NME from drained thermokarst lake basins (DTLB) and other select wetland/coastal tundra sites across the Arctic. The period of studies measurements for the studies observations are: a) mid-June – end of July b) June 12 – August 28, 2007, Fig 4 c) June 11 – August 25, 2011 d) upscaled chamber estimates, exact dates not specified, e) mean June 15 –August 31 2003-2006, f) July 5 – Aug 4, 2009.

| Site | Site Characteristics | NEE $g\ C\text{-}CO_2\ m^{-2}\ d^{-1}$ | NME $mg\ C\text{-}CH_4\ m^{-2}\ d^{-1}$ | Studies |
|---|---|---|---|---|
| Illisarvik | Young DTLB, Low & Tall Shrub/Grass/Wet Sedge | -1.5 | 8.7 | (this study) |
| Various DTLB, Barrow Peninsula, Alaska | Young DTLB, Wet Sedge Tundra | -1.1[b], -0.9[d], -0.8[c] | 18.4[a], 26.1[d], 44.0[c] | Zona et al. 2009[a] & 2010[b], Sturtevant and Oechel, 2013[c]; Lara et al. 2015[d] |
| | Medium DTLB, Wet Sedge Tundra | -0.7[b], -0.6[d], -0.4[c] | 27.0[d], 41.3[c] | |
| | Old DTLB, Wet Sedge Tundra | -1.0[b], -0.4[d], 0.1[c] | 24.2[d], 38.7[c] | |
| | Ancient DTLB, Wet Sedge Tundra | 0.4[d] | 21.7[d] | |

| Katyky, Indigirka lowlands, Siberia | Ancient DTLB, Dwarf-Shrub and Wet Sedge Tundra | -1.3[e] | 36.0[f] | Van der Molen et al. 2007[e], Budishchev et al. 2014[f] |
| --- | --- | --- | --- | --- |

**Figure 2a: Could you change the colour of the $T_a$ line? Red on orange is difficult to read.**

[Responses]

Agreed.  We have changed the bar colour, setting the orange to grey to make it easier to distinguish.

[Figure]

**Figure 3: I will leave this up to the author's discretion, but I wonder if this figure (and in fact, all figures) would benefit from having a plain white background. I find all the lines distracting. Especially when other lines are being used to annotate.**

[Responses]

We agree the grid is distracting for Figure 3 and removed it, but left grids in all the other figures.

[Figure]

**Figure 4c: Please use another three colours for Shrub. It is confusing that they are the same colour as VPD on the left-hand panel.**

[Responses]

We changed the colour scheme to address the concern.

[Figure]

**Figure 4 and 5: I think these figures would benefit from having a title for each panel – it was not clear to me initially the difference between Figure 5a and b. I think just putting VWC above left hand panel and T$_s$ above right hand panel, this would make it much clearer.**

[Responses]

We added subtitles to all panels in Figures 4 and 5.

**Tables: Caption should go above the Tables.**

[Responses]

Captions were all moved above Tables.

**I think a table including the dominant vegetation species for each class would be super useful for the reader.**

[Responses]

We updated Table 1 to be more specific and included additional information where known/applicable.

Table 1: Dominant species or landscape feature within the vegetation/cover classes. Unit codes correspond to the map Figure 1a.

| Unit Code | Vegetation Class | Dominant Species/Landscape feature |
|---|---|---|
| 1a | Shrub | *Salix alaxnesis* (Tall Willow) |
| 1b | Shrub | *Salix glauca* (Low Willow) |
| 1c | Shrub | *Alnus viridis* subsp. *crispa* (Alder) |
| 2a | Sedge Marsh | *Carex aquatilis* (Sedge) |
| 2b | Sedge Marsh | *Arctophila fulva* (Pendant Grass) |
| 3 | Grass Meadow | *Pocacea* spp. (Grasses), *Eriophorum angustifolium* (Cotton Grass) |
| 4a | Sparse Cover | Sparse Vegetation |
| 4b | Sparse Cover | Bare Ground |
| 5 | Ponds | *Hippuris vulgaris* (Mare's Tail), Open Water |
| 6a | Outside of Basin | Dwarf shrub tundra: *Salix* spp. & *Betula nana* (Birch) |
| 6b | Outside of Basin | Fen |
| 6c | Outside of Basin | Ocean |

**Reviewer # 2:**

**General comments**

**The paper "Vegetation Influence and Environmental Controls on Greenhouse Gas Fluxes from a Drained Thermokarst Lake in the Western Canadian Arctic" by June Skeeter et al. reports CO2 and CH4 flux measurements from a permafrost tundra site in Western Canada. Eddy covariance and chamber flux measurements were taken during the growing season 2016, and analysed accounting for the spatial variability of vegetation cover. Statistical gap-filling and an analysis of the environmental controls of the fluxes is performed using artificial neural networks. I think the chosen methods are properly applied and explained. The results are presented clearly and the conclusions are supported by the results. Also, the paper is very well written.**

**Given the remote and rather special site location, this study should be very valuable for the arctic carbon flux community. As the flux time series collected in your study may be used and referred to in future studies, it would be nice if you could present the time series in a more raw format than you do in Figure 2. For example, a plot of the 30-minute flux time series would help to understand the character of the data. This is also relevant, because I guess the performance and output of your NNs could be susceptible to noise or outliers in the EC time series.**

[Responses]

We changed Fig 2 to show the half hourly $F_{CO2}$ and $F_{CH4}$ observations, along with the $NEE_{NN}$ and $NME_{NN}$.

[Figure]

**Also, several of your results (cf. Line 221 and Line 342) are based on extrapolations into parts of the parameter space where the flux response could be governed by processed not captured in your NNs. Perhaps these statistical uncertainties could be discussed.**

[Responses]

We added a sentence at the end of section 2.5.1 discussing the impact of calculating ER by extrapolation and its impact on the confidence of ER estimates relative to NEE [Lines 265-267].  We refer the reader to Appendix A for details on the calculation of confidence intervals around NN outputs [Lines 501 -521].

This is a projection outside of the observed parameter space resulting in greater uncertainty and a wider confidence interval around $ER_{NN}$ than $NEE_{NN}$.  Calculation of confidence intervals for NN outputs is discussed in Appendix A

We also added a sentence to section 4.3 noting that projecting to Sedge = 100% is well outside of parameter space [Lines 433-434].

These are projections well beyond $F_{Clim}$ fractions observed so confidence in the specific values predicted is low.

**I understand there is little research from DTLB sites, but it would be good in your discussion to relate your findings to those from other tundra sites with (and without) thermokarst. In this discussion, it would be good to elaborate further on the peculiarities of the artificial draining performed at your site. Given the title of this paper, readers will probably expect more of these aspects discussed.**

[Responses]

We split the results into two separate sections "results" (Section 3) and "discussion" (Section 4).  In section 4.1 we discuss NEE and NME observations at Illisarvik relative to natural shrub vs. sedge-dominated DTLB to highlight the differences among these environments rather than attempt to fully contrast Illisarvik to a myriad of arctic tundra types/sites.   These comparisons are always challenging given different years, time periods within a year, instrumentation, and data presentation.   However, we believe we make a strong argument that shrub vs. sedge-dominated DTLB have the potential to differ and Illisarvik differs in particular from all other DTLB in its low methane emissions. (Table 4).  We now further highlight the important implications of vegetation succession on CO2 and CH4 fluxes at our site in the discussion section 4.3 "Future Trajectories".  This is one of the key messages associated with DTLBs – they undergo relatively rapid vegetation change over a number of decades that will influence their C budgets.

[revised manuscript text omitted]

**Specific comments**

**Line 16: "During the study period". Please be more specific here, because the upscaled average fluxes you mention in lines 18 and 20/21 don't tell much if you don't know the study period.**

[Responses]

Changed wording to "peak growing season" Line 16

**Line 24: Your abstract lacks a broader conclusion**

[Responses]

We added another sentence to make a broader conclusion about plant succession and Illisarvik's carbon balance, Lines 25-26

Presently, Illisarvik is a carbon sink during the peak growing season. However, these results suggest that rates of growing season CO2 and CH4 exchange rates may change as the basin's vegetation community continues to evolve.

**Line 100: Could the grazing have a measurable effect on e.g. NEE? It could be a point to add to your discussion.**

[Responses]

Good point, we added a few words to mention that grazing may have affected GHG Fluxes [Lines 121-122].

which may have affected greenhouse gas fluxes.

It is possible that grazing had some impact, but we cannot answer this based on the data collected. According to images from a fish eye camera mounted on the tripod (taken at 5-minute intervals), the animals spent about an hour gazing within the footprint of the eddy-covariance tower. In other areas of the basin where they stayed for longer, there was definitely a more significant impact. They were only spotted within the footprint the morning of July 12th. In addition

to the fish eye camera images, we were present at the field site during the full campaign and observed the reindeer's movements.

**Line 116: You discarded a sector because its flow could be disturbed by the tower.**

**But did you see this effect in any of your quality checks? Maybe it's not necessary to discard this data.**

[Responses]

It is standard practice to discard winds affected in the wake of the tower and sensor head. We have added a reference to Aubinet et al., 2012 to support this choice. During light winds, windspeeds can be reduced as much as 50% in the wake of a tower/instrument mount and turbulent eddies are artificially created, significantly violating the assumptions that go into eddy-covariance flux calculations. We oriented the tower such that this wind sector was the least frequent (according to climatology from Tuktoyaktuk). It only resulted in 6.7% (86 of 1279) half hourly observations being discarded.

**Line 146: Maybe be more specific about the Python modules you used, otherwise this sentence adds very little to the understanding of your analysis.**

[Responses]

We removed this portion of the manuscript. Most of the code was written specifically for the project by the first author, the footprint model of Kljun et al. 2015 is available in multiple programming languages, and we mention the python module for the neural networks on Line 243 and discuss the procedures in more in the appendix.

**Line 182: Shouldn't there be five times more vials than flux estimates, if you used 5 gas samples per flux measurement?**

[Responses]

Yes, that is correct. The sentence has been corrected [Lines 202-203].

After removal of spurious point measurements (72 vial samples were rejected out of 1135 vials), dc/dt was determined using three or more gas sample concentrations resulting in coefficients of determination that ranged from 0.71 to 0.99 The '681 flux measurements' referred to the three different gas fluxes each measurement produced ($CO_2$, $CH_4$, and $N_2O$). For clarity, we have removed this number (681) and we now only refer to $CO_2$ and $CH_4$ flux measurements (see also reply to reviewer #1)

**Line 292: "Random forest regression tree". Did you use only one decision tree, or the ensemble mean of several?**

[Responses]

It was the ensemble mean of 100 trees. We removed this from the text however. In retrospect, it was beyond the scope/point of the paper. Discussing the choices made for the random forest (RF) analysis would have required a new section in the methods. But since we didn't use it in the results (beyond this one comparison), this didn't seem necessary. Instead, we added a paragraph to the appendix [Lines 522-532] discussing why we RF weren't the best choice for this analysis and we added Figure A2 to support this.

Random forests (RF) are said to be among the best performing gap filling methods for NME (Kim et al., 2020). and it has been claimed that aggregating many regression trees in a RF prevents overfitting (Breiman, 2001;). We did not find this to be the case. Following the methods outlined in Kim et al. (2020): a RF with 400 trees and no restrictions on tree size fit FCH4 nearly perfectly (R2 = 0.98). Without considerable limitations on tree size, the RF will just learn

[Figure]

**Line 296: Maybe refer to an equation defining alpha.**

[Responses]

Alpha in this context is analogous to the minimum of the first derivative of the neural network output; which was calculated numerically. We added a new equation (Eq. 5) in section 2.5.1 [Line 258] to show a light response curve

$$NEE = \frac{1}{2c}\left(\alpha\text{PPFD} + \beta - \sqrt{(\alpha\text{PPFD} + \beta)^2 - 4\alpha\beta c\text{PPFD}}\right) + \text{ER}$$

and clarified section 3.3 to better describe this [Line 332-334].

The minimum values represent the peak light use efficiency and are analogous to α in eq. 5 (Fig 4b).

**Technical corrections**

**Line 75: Did you really mean 100 m, or maybe km?**

[Responses]

Yes, the antient basin, is just 100m to the south, it can be seen in Figure 1a (labeled 6b) and 1c in the top left of the drone image.

**Lines 302/303: Pa, with a capital P Line 310: "both"?**

[Responses]

Corrected, we also decided to use kPa instead

**Please check and correct the names of your references in the text, as several have spelling mistakes ("Whalen and Reedburgh", "Merbould", "Meyer-Smith")**

[Responses]

Thank you. We corrected these spelling mistakes.

**Figure 3b: Can you add a little bit of horizontal white space between the the Sedge plot and the rest? I think this could prevent confusion and make it clear that the y-axis for this box has a different scale**

[Responses]

We added the requested horizontal space and put "Sedge" into a separate subplot of the same figure.

[revised manuscript text omitted]

---

## Author Response (AR2)

**Response to the Editor: Vegetation Influence and Environmental Controls on Greenhouse Gas Fluxes from a Drained Thermokarst Lake in the Western Canadian Arctic**

The authors would like to thank the Editor for the helpful comments. Below is a point by point response to the Editor's comments. The comments are in bold, responses are noted below each comment and shown in plain text. Changes to the manuscript are in red. Following that, is a marked-up version of the revised manuscript showing all changes made.

**What about winter contribution and indeed the rates may change but in which anticipated direction?**

[Response]

Winter contributions are well beyond the scope of this study so we don't feel comfortable speaking to that in the Abstract. There are likely significant methane emissions during the fall when the active layer is freezing back up (e.g. Zona et al. 2016), but the degree to which this would offset growing season carbon uptake is unknown this is a topic that needs to be addressed by further studies in the future. However, the harsh conditions, remoteness and low light levels present a major logistical challenge when operating EC systems at sites like Illisarvik over the winter.

In regards to the direction of anticipated changes during the growing season, we speak to this in section 4.3 Future Trajectories [Lines 421-441]. As these results are extrapolations to unknown future vegetation conditions, we didn't feel comfortable making a definitive statement on the direction of changes in the abstract. If the basin's sedge cover increases and shrub cover decreases, we expect NME to increase significantly. If the opposite occurs, we anticipate NME would be similar to current observations. This is highlighted in Figure 5a.

There is less certainty in regards to NEE as vegetation type wasn't the dominate control identified by the model. Figure 4 c. shows the model suggests that ER and NEE will change with an increase or decrease in shrub cover, but the direction of these changes will be governed by changes in soil moisture. Under wetter conditions at 100% shrub cover to decrease NEE considerably by increasing ER. While the inverse is true under drier conditions. At 0% shrub cover, the difference is less dramatic, but the model suggest higher VWC decreases ER.

To make these projections clearer to the reader, we have added references to fig 4c and 5a in section 4.3.

Figure 5a shows that with extrapolations to full Sedge cover ($F_{Sedge}$ = 100%), NME would be similar to values on the Barrow Peninsula (Zona et al., 2009). If the basin instead transitions into a shrub dominated DTLB similar to those of Old Crow Flats, Yukon (Lantz et al., 2015), $NME_{NN}$ would remain similar to current levels meaning the basin would remain a weak source of $CH_4$. These are projections well beyond $F_{Clim}$ fractions observed so confidence in the specific values predicted is low.

The effects of changing shrub/sedge cover on Illisarvik's growing season NEE are less straightforward than on NME. Partly because Shrub cover had less overall influence on $NEE_{NN}$. Figure 4c. shows the model suggests ER decreases and NEE increases with increasing shrub coverage when soils are slightly drier, but has the opposite effect under wetter conditions.

**repetitive please adjust**

[Response]

We changed wording to:

Thermokarst lakes are a prominent landscape feature of the Western Canadian Arctic (Mackay, 1999; Marsh et al, 2009; Lantz & Turner, 2015). These lakes drain, sometimes catastrophically, forming drained thermokarst lake basins (DTLB) via bank overflow, ice wedge erosion, coastal erosion, and stream migration (Billings and Peterson, 1980; Mackay, 1999).

**your results tell another story**

[Response]

We don't feel that our results refute this point. These studies are referencing longer term shifts in NEE which are related to changes in ER due to changing soil moisture and vegetation composition associated with climate change. Yes, NEE is most directly controlled on the half hourly time scale by PPFD directly governing GPP. But our model shows that changes in shrub cover and soil moisture influence ER which in turn influences NEE. These types of changes can have a sustained influence over longer timescales.

**Subscript**

[Response]

Fixed

**compared to where?**

We added "than Alaskan DTLB" to make this comparison more clear

**please add seasonal here.**

[Response]

We added "during the growing season"

**custom-made**

[Response]

Fixed

**remove the ?**

[Response]

Fixed

**Chamber**

[Response]

Fixed

**reference**

We changed below to "in section 2.5.1." to make the reference clearer.

**and what was the result of this exercise?**

The vials did indeed maintain their integrity.    We've changed the sentence to:

The integrity of the vials through shipping, storage and analysis was confirmed using a subset filled with helium before the field season began

**Why? bubbles or leakage?**

Bubbles, leakage, or other errors.  We've updated this section to be clearer.  Also, further changes are needed to that sentence as we were only reporting the $CO_2$ $R^2$ results.  $CH_4$ $R^2$ are never as nice and clean so added that information to be more explicit.

Visual inspection of the linear trend of gas concentrations (dc/dt) was used to identify and remove spurious point measurements associated with analysis errors, leaking chambers (isolated decreases in concentration) and contamination or ebullition events (isolated increases in concentration) (0.3%, 0.7%, and 2.0% of $CO_2$ samples and 2.1%, 0.5%, and 1.1% of $CH_4$ samples, respectively).  In all flux measurements, at least three or more gas samples remained so that dc/dt and its coefficient of determination ($R^2$) were determined using least squares linear regression.  We did not use $R^2$ as an additional quality control criterion as many of our $CH_4$ fluxes were near zero and tended to have low $R^2$ values due to only small variations in the point sample concentrations (see also Clark et al., 2020).  40% and 32% of the 227 $CH_4$ flux measurements and 97% and 92% of the 227 $CO_2$ flux measurements had $R^2$ over 0.80 and 0.90, respectively.

**Rh**

[Response]

Fixed

**(Aubinet...)**

[Response]

We are unsure what this comment means.  The name isn't spelled wrong and the reference appears to be correct?

**did not have**

[Response]

Fixed

**-**

[Response]

Fixed

**this is unclear. you measured NME and then you estimate NME with NMENN? kindly clarify**

[Response]

We revised the sentence to be clearer

$NME_{NN}$ ($r^2 = 0.62$) was estimated using five factors: $F_{Sedge}$, $F_{Shrub}$ , $VWC$, $T_S$, and $U$.

**sometimes you state NN NME and then NME NN - kindly harmonize**

[Response]

We had changed the notation from the original submission.  But there were some typos that carried over from the original.  We've revised the notations to be consistent.

**DTLBs**

We updated all plural instances of DTLB to reflect this suggestion

**gap filled EC estimates**

[Response]

Fixed

**Wording: no year round studies have yet to be published – (remove the no at the beginning)**

[Response]

We changed the wording from "have yet to be" to "have been".  We also moved DTLB up in the sentence to make it clearer we are saying no year round DTLB studies exist.

To our knowledge, only few winter season (e.g. Zona et al. 2016) and no year-round studies of DTLB NEE and NME have been published to help evaluate the factors influencing carbon losses through the non-growing season months.

**Also are you sure there arent any year round measurements - even not from Siberia?**

[Response]

To our knowledge, no there are not any year-round DTLB studies.  In Siberia, the only DTLB that has been studied to our knowledge is Katyk, which has only been studied during the growing season.  The remoteness, harsh weather, and lack of sunlight make powering EC stations over winter in the arctic very difficult.

**Response to Reviews: Vegetation Influence and Environmental Controls on Greenhouse Gas Fluxes from a Drained Thermokarst Lake in the Western Canadian Arctic**

The authors would like to thank the reviewers for their valuable comments and criticisms. Point by point responses to both reviewers are included in this document. Reviewer comments are in bold, responses are noted below each

comment and shown in plain text. Changes to the manuscript are in red. Following that, is a marked-up version of the revised manuscript showing all changes made.

**Reviewer #1**

**This paper by Skeeter et al. looked at vegetation and environmental conditions influencing greenhouse gas exchange in a drained lake basin in the Western Canadian Arctic. I enjoyed the opportunity to review this paper and I thank the authors for what is a well written paper overall (but with some tweaks needed). Given the lack of studies outside of the Barrow Peninsula, is a worthy addition to the literature.**

**I will add however, I apologise, I am not an expert on the eddy covariance data cleaning, gapfilling and analyses and I am therefore unable to comment fully on those sections.**

**Introduction: I find this section well written if rather short. I think a little more context could be given for the reader. You could include more information from non-DLB work but still relevant arctic tundra literature.**
[Responses]
We added more context on why thermokarst landscapes are important (spatial extent, significant soil carbon storage) to the first paragraph [Lines 32-34].

Lake thermokarst landscapes are widespread in poorly drained, sedimentary permafrost lowlands with excess ground ice volume and constitute about a third of all thermokarst area (French, 2013; Olefeldt et al., 2016).

Further we added another paragraph on Arctic carbon budgets [Lines 40-49]. Including a review of pan-arctic NEE chamber studies by Virkkala et al. 2017.

Net ecosystem exchange (NEE), ecosystem respiration (ER) and gross primary productivity (GPP), where NEE=ER-GPP are lower in the Arctic than warmer regions but have significant seasonal cycles and variability between vegetation types (Virkkala et al., 2018). Future trajectories in NEE will in large part be governed by ER (Biasi et al., 2008; Cahoon et al.,2012). Dominant vegetation types in the Western Canadian Arctic are erect-shrub tundra and wetlands (Walker et al., 2005). Growing season NEE is typically negative across these units throughout the Arctic indicating a net $CO_2$ sink as GPP exceeds ER in part due to cold and/or anoxic soil conditions (Virkkala et al., 2018; Lafleur et al., 2012). Annual NEE can be positive or negative with large variation in GPP linked to annual weather variability (Virkkala et al., 2018, McGuire et al., 2009). Arctic net methane exchange (NME) is positive because wetland areas are strong methane ($CH_4$) sources while upland areas with better drainage can be net sinks (Whalen and Reeburgh, 1990; McGuire et al., 2009; Sturtevant and Oechel, 2013).

**Line 54: NEE should be ER – GPP.**
[Responses]

We corrected the equation to NEE = ER – GPP

**Line 85: Can you include somewhere the dominant species found in each vegetation class. It would be interesting and useful to know what sort of sedge dominated the sedge class – is it Carex aquatilis or Eriophorum angustifolium for example?**

[Responses]

We added a sentence to mention the specific species [Lines 90-92].

Current vegetation at Illisarvik is diverse relative to the dwarf-shrub tundra of the surrounding uplands (Table 1); the basin hosts a mix of woody shrubs (Salix spp., Betula spp., & Alnus spp), wetland vegetation (Carex aquatilis, Arctophila fulva, etc.), and various grasses (Pocacea spp.) (Wilson et al. 2019).

This information was partly available in Table 1.  We made the table more detailed to include the dominant species present within each class/subclass where known/applicable (see comment below).

*Line 100: Completely an aside but COOL!*

* Yes, it was an amazing thing to see.

**Section 2.3: I would remove any mention of N2O – you don't present the data, so it is unnecessary.**

[Responses]

We removed mentions of $N_2O$ throughout the manuscript.

**Line 150: Can you be more explicit with how many collars were used? 2 per site – 10 sites total? I know the main focus of this paper is not the collars, but I'm not sure if a replicate of 2 per vegetation type over an 11-day period is very representative.**

[Responses]

We updated the text to be more specific about numbers of replications per vegetation type [Lines 176-177].

There were three replicates (six collars) for the Shrub class, two for the Sedge, Grass, and Upland tundra, and no replicates for the Sparse class.

There were 19 collars and 10 sites. The bare ground site only had 1 collar, thus making 2 per sites for the rest. The number of collars that could be shipped in via helicopter were limited and there was a high amount of heterogeneity in soil and vegetation characteristics within the basin. The chamber study was designed to better understand the relationship between soil properties and carbon loss in a situation where permafrost had aggraded within the lake bed to potentially protect 'old' carbon from mineralization and accumulate 'new' carbon since the lake drained.  We expected saturated soils (where there were wet sedges) to have higher organic carbon accumulation and be dominated by anaerobic respiration processes, which was interesting to us, therefore we chose two different wet sites populated

by sedges. We also focused on different statures of will (low, tall and dense) as we expected different amounts of snow accumulation and different impacts on permafrost at these sites.

**Line 152: How soon after installation were the collars fluxed for the first time?**

[Responses]

Collars were installed on July 11[th] and first set of measurements was taken on July 12[th], so about 24h.

**Line 154: Why not use a clear chamber so you could get GPP then cover with a dark sleeve in order to get ER?**

[Responses]

We appreciate the comment. The chosen chambers were not designed for NEE measurements. Although measurements of GPP would have been informative, logistics limited us to use the existing collars, and the number of measurements we could make.

**Section 2.4 (and subsequently Appendix A): Unfortunately, I do not have the expertise in these methods so I do not feel comfortable commenting on it in a reviewer context.**

**Section 3: I think it would be better to separate this section out into Results and Discussion rather than combine them. As it stands, it's quite hard to follow.**

We have added a discussion (section 4) and rewrote the results to solely contain the objectively retrieved data, so we hope the manuscript to be more straightforward and easier to follow with separate "Results" and "Discussion" sections.

[revised manuscript text omitted]

**Line 241: You only mentioned thaw depth twice? Why not measure it on each day chambers were used?**

[Responses]

Thaw depth tends to increase over time but at different rates at different locations within the basin as a result of varying soil and surface properties. We measured thaw depth at the start and end of the measurement period to highlight these differences rather than develop a variable that could be related to the fluxes. In past studies we found that day to day variations in respiration correlate best with near surface soil temperature and moisture rather than thaw depth while spatial variations in average fluxes can sometimes correlate to max thaw depth. In the revised manuscript we refer to thaw depth more and use it to compare Illisarvik to Katyk Line 384

ER at Illisarvik was greater than the ER observed at both the young wet-sedge DTLB in Barrow (Zona et al., 2010) and at the shrub/wet sedge DTLB at Katyk where thaw depth was much shallower (45 to >100 cm at Illisarvik vs. 25 to 40 cm at Katyk; van der Molen et al. 2007).

**Section 3.1: I think this section needs an overhaul unfortunately. Many of the sentences do not make sense in their current format. For example: Lines 254-257: 'NEE was greater than (ie. Less carbon uptake) 255 EC observations of from four wetter, sedge dominated DTLB, where peak season NEE was -2.5 g C-CO2 m-2 d-1, ER (1.5 g C-CO2 m-2 d-1) was lower than at Illisarvik while GPP (4.0 g C-CO2 m-2 d-1) was slightly higher (Zona et al., 2010).'**

[Responses]

The entire section has been rewritten and the references to other work were moved out from the results to the discussion section. See previous comment response.

**I also think it might be useful to separate the EC results and the chamber results into subsections. By referring your measured values to other studies in the results, it makes it hard to follow for the reader.**

[Responses]

We agree and have separated the EC and chamber results into Sections 3.1 and 3.2. These changes are shown in the above comment about section 3.

**Further, by combining the results, there is a lack of discussing the results (for example, it feels like only section 3.4 is really doing this). It sadly reads as a lot of results statements and then suddenly we are at the conclusion.**

[Responses]

We added a separate "discussion" section (Section 4). These changes are shown in the above comment about section 3.

**Line 273: Why compare methane to ER here?**

[Responses]

We thought it is relevant to contextualize the differences. Methane emissions are far more spatially variable than ecosystem respiration. We revised the sentence to get rid of the comparison of the magnitude because that is less relevant, but left the point about spatial variability being enhanced [Line 316]. We think it is an important finding to show that NME is more influenced by spatial heterogeneity than ER.

NME was more variable between vegetation classes than ER (Fig 3b & c).

**Line 286: Although discrepancies do occur between upscaling chamber measurements and EC measurements – some studies have done it successfully and I think it would be good to include here as a caveat;**

- **Budischev et al. 2014: Evaluation of a plot-scale methane emission model using eddy covariance observations and footprint modelling. Biogeosciences 11. 4651-4664**
- **Zhang et al. 2012: Upscaling methane fluxes from closed chambers to eddy covariance based on a permafrost integrated model. Global Change Biology, 18, 1428-1440.**
- **Davidson et al. 2017. Upscaling CH4 fluxes using high-resolution imagery in Arctic Tundra Ecosystems. Remote Sensing, 9, 1227; doi:10.3390/rs9121227**
- **Sachs et al. 2010 Environmental controls on CH4 emissions from polygonal tundra on the microsite scale in the Lena river delta, Siberia. Global Change Biology, 16, 3096-3110**

[Responses]

We reviewed this literature and added it to the newly separated "discussion" section [Lines 417-418].

Others have been more successful, yielding upscaled chamber NME fluxes within 10% of EC observations (Zhang et al., 2012; Budishchev et al., 2014; Davidson et al., 2017).

We also decided to use the footprint weighted upscaling method discussed in Budishchev et al. (2014) for the chamber upscaling [Line 207-208], but it did not make an appreciable difference in the upscaled chamber ER or NME.

Chamber fluxes of ER were upscaled from the plot scale (individual chamber) to the footprint scale using the footprint weighted average method and to the basin scale using the area weighted average method (Budishchev et al., 2014).

**I think more discussion of the results in the context of other GHG literature from other tundra ecosites would be useful. Although this study is focused on drained lake basins, the results are comparable to wet-sedge**

**dominated tundra landscapes. I feel this would be a good addition and strengthen what is already a useful paper.**

[Responses]

We added section 4.1 (see above comment) where we discuss NEE and NME observations at Illisarvik relative to natural shrub vs. sedge-dominated DTLB to highlight the differences among these environments rather than attempt to fully contrast Illisarvik to a myriad of arctic tundra types/sites. These comparisons are always challenging given different years, time periods within a year, instrumentation, and data presentation. However, we believe we make a strong argument that shrub vs. sedge-dominated DTLB have the potential to differ and Illisarvik differs in particular from all other DTLB in its low methane emissions. (Table 4). We now further highlight the important implications of vegetation succession on CO2 and CH4 fluxes at our site in the discussion section 4.3 "Future Trajectories". This is one of the key messages associated with DTLBs – they undergo relatively rapid vegetation change over a number of decades that will influence their C budgets.

Table 4: Growing season (gs) daily range in eddy covariance-derived NEE and NME from drained thermokarst lake basins (DTLB) and other select wetland/coastal tundra sites across the Arctic. The period of studies measurements for the studies observations are: a) mid-June – end of July b) June 12 – August 28, 2007, Fig 4 c) June 11 – August 25, 2011 d) upscaled chamber estimates, exact dates not specified, e) mean June 15 –August 31 2003-2006, f) July 5 – Aug 4, 2009.

| Site | Site Characteristics | NEE $g\ C\text{-}CO_2\ m^{-2}\ d^{-1}$ | NME $mg\ C\text{-}CH_4\ m^{-2}\ d^{-1}$ | Studies |
|---|---|---|---|---|
| Illisarvik | Young DTLB, Low & Tall Shrub/Grass/Wet Sedge | -1.5 | 8.7 | (this study) |
| Various DTLB, Barrow Peninsula, Alaska | Young DTLB, Wet Sedge Tundra | -1.1[b], -0.9[d], -0.8[c] | 18.4[a], 26.1[d], 44.0[c] | Zona et al. 2009[a] & 2010[b], Sturtevant and Oechel, 2013[c]; Lara et al. 2015[d] |
| | Medium DTLB, Wet Sedge Tundra | -0.7[b], -0.6[d], -0.4[c] | 27.0[d], 41.3[c] | |
| | Old DTLB, Wet Sedge Tundra | -1.0[b], -0.4[d], 0.1[c] | 24.2[d], 38.7[c] | |
| | Ancient DTLB, Wet Sedge Tundra | 0.4[d] | 21.7[d] | |
| Katyky, Indigirka lowlands, Siberia | Ancient DTLB, Dwarf-Shrub and Wet Sedge Tundra | -1.3[e] | 36.0[f] | Van der Molen et al. 2007[e], Budishchev et al. 2014[f] |

**Figure 2a: Could you change the colour of the T$_a$ line? Red on orange is difficult to read.**

[Responses]

Agreed.  We have changed the bar colour, setting the orange to grey to make it easier to distinguish.

[Figure]

**Figure 3: I will leave this up to the author's discretion, but I wonder if this figure (and in fact, all figures) would benefit from having a plain white background. I find all the lines distracting. Especially when other lines are being used to annotate.**

[Responses]

We agree the grid is distracting for Figure 3 and removed it, but left grids in all the other figures.

[Figure]

**Figure 4c: Please use another three colours for Shrub. It is confusing that they are the same colour as VPD on the left-hand panel.**

[Responses]

We changed the colour scheme to address the concern.

[Figure]

**Figure 4 and 5: I think these figures would benefit from having a title for each panel – it was not clear to me initially the difference between Figure 5a and b. I think just putting VWC above left hand panel and T$_s$ above right hand panel, this would make it much clearer.**

[Responses]

We added subtitles to all panels in Figures 4 and 5.

**Tables: Caption should go above the Tables.**

[Responses]

Captions were all moved above Tables.

**I think a table including the dominant vegetation species for each class would be super useful for the reader.**

[Responses]

We updated Table 1 to be more specific and included additional information where known/applicable.

Table 1: Dominant species or landscape feature within the vegetation/cover classes.  Unit codes correspond to the map Figure 1a.

| Unit Code | Vegetation Class | Dominant Species/Landscape feature |
|---|---|---|
| 1a | Shrub | *Salix alaxnesis* (Tall Willow) |
| 1b | Shrub | *Salix glauca* (Low Willow) |
| 1c | Shrub | *Alnus viridis* subsp. *crispa* (Alder) |
| 2a | Sedge Marsh | *Carex aquatilis* (Sedge) |
| 2b | Sedge Marsh | *Arctophila fulva* (Pendant Grass) |
| 3 | Grass Meadow | *Pocacea* spp. (Grasses), *Eriophorum angustifolium* (Cotton Grass) |
| 4a | Sparse Cover | Sparse Vegetation |
| 4b | Sparse Cover | Bare Ground |
| 5 | Ponds | *Hippuris vulgaris* (Mare's Tail), Open Water |
| 6a | Outside of Basin | Dwarf shrub tundra: *Salix* spp. & *Betula nana* (Birch) |
| 6b | Outside of Basin | Fen |
| 6c | Outside of Basin | Ocean |

**Reviewer # 2:**

**General comments**

**The paper "Vegetation Influence and Environmental Controls on Greenhouse Gas Fluxes from a Drained Thermokarst Lake in the Western Canadian Arctic" by June Skeeter et al. reports CO2 and CH4 flux measurements from a permafrost tundra site in Western Canada. Eddy covariance and chamber flux measurements were taken during the growing season 2016, and analysed accounting for the spatial variability of vegetation cover. Statistical gap-filling and an analysis of the environmental controls of the fluxes is performed using artificial neural networks. I think the chosen methods are properly applied and explained. The results are presented clearly and the conclusions are supported by the results. Also, the paper is very well written.**

**Given the remote and rather special site location, this study should be very valuable for the arctic carbon flux community. As the flux time series collected in your study may be used and referred to in future studies, it would be nice if you could present the time series in a more raw format than you do in Figure 2. For example, a plot of the 30-minute flux time series would help to understand the character of the data. This is also relevant, because I guess the performance and output of your NNs could be susceptible to noise or outliers in the EC time series.**

[Responses]

We changed Fig 2 to show the half hourly $F_{CO2}$ and $F_{CH4}$ observations, along with the $NEE_{NN}$ and $NME_{NN}$.

[Figure]

**Also, several of your results (cf. Line 221 and Line 342) are based on extrapolations into parts of the parameter space where the flux response could be governed by processed not captured in your NNs. Perhaps these statistical uncertainties could be discussed.**

[Responses]

We added a sentence at the end of section 2.5.1 discussing the impact of calculating ER by extrapolation and its impact on the confidence of ER estimates relative to NEE [Lines 265-267]. We refer the reader to Appendix A for details on the calculation of confidence intervals around NN outputs [Lines 501 -521].

This is a projection outside of the observed parameter space resulting in greater uncertainty and a wider confidence interval around $ER_{NN}$ than $NEE_{NN}$. Calculation of confidence intervals for NN outputs is discussed in Appendix A

We also added a sentence to section 4.3 noting that projecting to Sedge = 100% is well outside of parameter space [Lines 433-434].

These are projections well beyond $F_{Clim}$ fractions observed so confidence in the specific values predicted is low.

**I understand there is little research from DTLB sites, but it would be good in your discussion to relate your findings to those from other tundra sites with (and without) thermokarst. In this discussion, it would be good to elaborate further on the peculiarities of the artificial draining performed at your site. Given the title of this paper, readers will probably expect more of these aspects discussed.**

[Responses]

We split the results into two separate sections "results" (Section 3) and "discussion" (Section 4). In section 4.1 we discuss NEE and NME observations at Illisarvik relative to natural shrub vs. sedge-dominated DTLB to highlight the differences among these environments rather than attempt to fully contrast Illisarvik to a myriad of arctic tundra types/sites. These comparisons are always challenging given different years, time periods within a year, instrumentation, and data presentation. However, we believe we make a strong argument that shrub vs. sedge-dominated DTLB have the potential to differ and Illisarvik differs in particular from all other DTLB in its low methane emissions. (Table 4). We now further highlight the important implications of vegetation succession on CO2 and CH4 fluxes at our site in the discussion section 4.3 "Future Trajectories". This is one of the key messages associated with DTLBs – they undergo relatively rapid vegetation change over a number of decades that will influence their C budgets.

[revised manuscript text omitted]

**Specific comments**

**Line 16: "During the study period". Please be more specific here, because the upscaled average fluxes you mention in lines 18 and 20/21 don't tell much if you don't know the study period.**

[Responses]

Changed wording to "peak growing season" Line 16

**Line 24: Your abstract lacks a broader conclusion**

[Responses]

We added another sentence to make a broader conclusion about plant succession and Illisarvik's carbon balance, Lines 25-26

Presently, Illisarvik is a carbon sink during the peak growing season. However, these results suggest that rates of growing season CO2 and CH4 exchange rates may change as the basin's vegetation community continues to evolve.

**Line 100: Could the grazing have a measurable effect on e.g. NEE? It could be a point to add to your discussion.**

[Responses]

Good point, we added a few words to mention that grazing may have affected GHG Fluxes [Lines 121-122].

which may have affected greenhouse gas fluxes.

It is possible that grazing had some impact, but we cannot answer this based on the data collected. According to images from a fish eye camera mounted on the tripod (taken at 5-minute intervals), the animals spent about an hour gazing within the footprint of the eddy-covariance tower.  In other areas of the basin where they stayed for longer, there was definitely a more significant impact.  They were only spotted within the footprint the morning of July 12th.  In addition

to the fish eye camera images, we were present at the field site during the full campaign and observed the reindeer's movements.

**Line 116: You discarded a sector because its flow could be disturbed by the tower.**

**But did you see this effect in any of your quality checks? Maybe it's not necessary to discard this data.**

[Responses]

It is standard practice to discard winds affected in the wake of the tower and sensor head. We have added a reference to Aubinet et al., 2012 to support this choice. During light winds, windspeeds can be reduced as much as 50% in the wake of a tower/instrument mount and turbulent eddies are artificially created, significantly violating the assumptions that go into eddy-covariance flux calculations. We oriented the tower such that this wind sector was the least frequent (according to climatology from Tuktoyaktuk). It only resulted in 6.7% (86 of 1279) half hourly observations being discarded.

**Line 146: Maybe be more specific about the Python modules you used, otherwise this sentence adds very little to the understanding of your analysis.**

[Responses]

We removed this portion of the manuscript. Most of the code was written specifically for the project by the first author, the footprint model of Kljun et al. 2015 is available in multiple programming languages, and we mention the python module for the neural networks on Line 243 and discuss the procedures in more in the appendix.

**Line 182: Shouldn't there be five times more vials than flux estimates, if you used 5 gas samples per flux measurement?**

[Responses]

Yes, that is correct. The sentence has been corrected [Lines 202-203].

After removal of spurious point measurements (72 vial samples were rejected out of 1135 vials), dc/dt was determined using three or more gas sample concentrations resulting in coefficients of determination that ranged from 0.71 to 0.99 The '681 flux measurements' referred to the three different gas fluxes each measurement produced ($CO_2$, $CH_4$, and $N_2O$). For clarity, we have removed this number (681) and we now only refer to $CO_2$ and $CH_4$ flux measurements (see also reply to reviewer #1)

**Line 292: "Random forest regression tree". Did you use only one decision tree, or the ensemble mean of several?**

[Responses]

It was the ensemble mean of 100 trees. We removed this from the text however. In retrospect, it was beyond the scope/point of the paper. Discussing the choices made for the random forest (RF) analysis would have required a new section in the methods. But since we didn't use it in the results (beyond this one comparison), this didn't seem necessary. Instead, we added a paragraph to the appendix [Lines 522-532] discussing why we RF weren't the best choice for this analysis and we added Figure A2 to support this.

Random forests (RF) are said to be among the best performing gap filling methods for NME (Kim et al., 2020). and it has been claimed that aggregating many regression trees in a RF prevents overfitting (Breiman, 2001;). We did not find this to be the case. Following the methods outlined in Kim et al. (2020): a RF with 400 trees and no restrictions on tree size fit FCH4 nearly perfectly (R2 = 0.98). Without considerable limitations on tree size, the RF will just learn

the dataset rather than the relationships present. It is our view that this tree is extremely overfit, as highlighted by the example in Figure A2. Further, RF do not allow for straightforward visualization functional relationships in a dataset. Plotting FCH4 against VWC, which is the dominant environmental control identified does not reveal a meaningful relationship like Figure 5 a & c. You can look at an individual decision tree within the RF, but those are difficult to interpret beyond the first few splits, and each tree will be different. Lastly, RF are incapable of projecting beyond the parameter space observed which limited their applicability for this study (Fig A2). This presents an issue because may gaps in EC data arise from data filtering (e.g. clear calm nights, precipitation events) and are by definition outside the parameter space observed.

[Figure]

**Line 296: Maybe refer to an equation defining alpha.**

[Responses]

Alpha in this context is analogous to the minimum of the first derivative of the neural network output; which was calculated numerically. We added a new equation (Eq. 5) in section 2.5.1 [Line 258] to show a light response curve

$$NEE = \frac{1}{2c}\left(\alpha\text{PPFD} + \beta - \sqrt{(\alpha\text{PPFD} + \beta)^2 - 4\alpha\beta c\text{PPFD}}\right) + \text{ER}$$

and clarified section 3.3 to better describe this [Line 332-334].

The minimum values represent the peak light use efficiency and are analogous to α in eq. 5 (Fig 4b).

**Technical corrections**

**Line 75: Did you really mean 100 m, or maybe km?**

[Responses]

Yes, the antient basin, is just 100m to the south, it can be seen in Figure 1a (labeled 6b) and 1c in the top left of the drone image.

**Lines 302/303: Pa, with a capital P Line 310: "both"?**

[Responses]

Corrected, we also decided to use kPa instead

**Please check and correct the names of your references in the text, as several have spelling mistakes ("Whalen and Reedburgh", "Merbould", "Meyer-Smith")**

[Responses]

Thank you. We corrected these spelling mistakes.

**Figure 3b: Can you add a little bit of horizontal white space between the the Sedge plot and the rest? I think this could prevent confusion and make it clear that the y-axis for this box has a different scale**

[Responses]

We added the requested horizontal space and put "Sedge" into a separate subplot of the same figure.

[revised manuscript text omitted]